# Evaluating the Antitumor Potential of Cannabichromene, Cannabigerol, and Related Compounds from *Cannabis sativa* and *Piper nigrum* Against Malignant Glioma: An In Silico to In Vitro Approach

**DOI:** 10.3390/ijms26125688

**Published:** 2025-06-13

**Authors:** Andrés David Turizo Smith, Nicolás Montoya Moreno, Josefa Antonia Rodríguez-García, Juan Camilo Marín-Loaiza, Gonzalo Arboleda Bustos

**Affiliations:** 1Departamento de Patología, Facultad de Medicina, Universidad Nacional de Colombia, Bogotá 111221, Colombia; gharboledab@unal.edu.co; 2Departamento de Química, Facultad de Ciencias, Universidad Nacional de Colombia, Bogotá 111221, Colombia; nicmontoyamor@unal.edu.co; 3Grupo de Investigación en Biología del Cáncer, Instituto Nacional de Cancerología, Bogotá 111221, Colombia; jrodriguez@cancer.gov.co; 4Departamento de Farmacia, Facultad de Ciencias, Universidad Nacional de Colombia, Bogotá 111221, Colombia; jcmarinlo@unal.edu.co

**Keywords:** GPR55, PINK1, glioblastoma, *Cannabis sativa*, *Piper nigrum*, cannabichromene, cannabigerol, cannabidiolic acid (CBDA), piperine, cannabidiol, U87MG, T98G, CCF-STTG1

## Abstract

Malignant gliomas, including glioblastoma multiforme (GBM), are highly aggressive brain tumors with a poor prognosis and limited treatment options. This study investigates the antitumor potential of bioactive compounds derived from *Cannabis sativa* and *Piper nigrum* using molecular docking, cell viability assays, and transcriptomic and expression analyses from public databases in humans and cell lines. Cannabichromene (CBC), cannabigerol (CBG), cannabidiol (CBD), and Piper nigrum derivates exhibited strong binding affinities relative to glioblastoma-associated targets GPR55 and PINK1. In vitro analyses demonstrated their cytotoxic effects on glioblastoma cell lines (U87MG, T98G, and CCF-STTG1), as well as on neuroblastoma (SH-SY5Y) and oligodendroglial (MO3.13) cell lines, revealing interactions among these compounds. The differential expression of GPR55 and PINK1 in tumor versus normal tissues further supports their potential as biomarkers and therapeutic targets. These findings provide a basis for the development of novel therapies and suggest unexplored molecular pathways for the treatment of malignant glioma.

## 1. Introduction

Gliomas are tumors originating from glial cells, and they are the most common type of primary brain tumors. High-grade gliomas represent the most aggressive form of these tumors [1]. Malignant gliomas present significant treatment challenges, with little improvement in the median survival time, which remains around 15 months over the past 30 years [2]. Treatment options currently include surgery, radiotherapy (RT), and chemotherapy [3]. However, new therapies targeting the suppression of new blood vessel formation (antiangiogenic treatments) [2] and immunotherapy [4] are emerging as promising strategies to overcome resistance to traditional treatments in GBM [2]. Additionally, nanoparticle-mediated therapies leverage the unique properties of nanoparticles (NPs) to improve drug delivery, enhance targeting, and reduce side effects. A key advantage of nanoparticles is their ability to cross the blood–brain barrier (BBB), a major obstacle in GBM treatment [5]. Despite all these efforts, the results remain discouraging, and effective therapeutic alternatives continue to be an unmet medical need [6].

GBM is the most aggressive and lethal primary brain tumor, known for its rapid progression, therapeutic resistance, and poor prognosis. The incidence rate of glioblastoma has risen over the years, ranging from 0.59 to 5 per 100,000 individuals [7]. Despite advancements in neurosurgery, radiotherapy, and chemotherapy, the median survival for GBM patients remains dismal, typically less than 15 months, with only 5% of diagnosed individuals surviving more than five years. This is due to the tumor’s infiltrative nature, molecular heterogeneity, and resistance mechanisms driven by the tumor microenvironment and glioblastoma stem cells (GSCs). These factors highlight the critical need for innovative therapeutic strategies targeting GBM’s molecular vulnerabilities [1,8].

The growing field of cannabinoid research has introduced novel avenues for cancer therapy. Compounds derived from *Cannabis sativa* have demonstrated potential in modulating key oncogenic pathways through their interactions with cannabinoid receptors (CB1R, CB2R, and GPR55), which form a part of the endocannabinoid system (ECS) [2,9]. Among these compounds, cannabichromene (CBC), cannabigerol (CBG), and cannabidiol (CBD) exhibit anti-inflammatory, pro-apoptotic, and anti-proliferative properties in various cancer models [2,3,4,5,6,7,9,10,11,12,13,14]. Similarly, secondary metabolites from *Piper nigrum*, such as P and its derivatives, have demonstrated antitumor properties, which are attributed to their activity as TRPV1 agonists and their ability to modulate oxidative stress and apoptosis [8,9,10,11,12,15,16,17,18,19]. However, the combined effects of cannabinoids and *Piper nigrum* compounds in the context of GBM remain largely unexplored. Table 1 summarizes some of the key components found in these plants and used in this study.

The selection of *Cannabis sativa*-derived cannabinoids and *Piper nigrum* compounds for GBM therapy is based on their ability to modulate key oncogenic pathways and their potential for synergistic interactions. *Cannabis sativa*, traditionally used as an herbal remedy for centuries, is the primary source of phytocannabinoids, which interact with the ECS—a network of receptors, endogenous ligands, and enzymes that regulate various physiological and pathological processes, including cancer [20]. Phytocannabinoids with other bioactive compounds can modulate ECS components and other cellular pathways, thereby influencing tumor progression and metastasis [9]. Given the growing body of in vitro and in vivo evidence demonstrating the tumor-inhibitory and antimetastatic effects of cannabinoids, they are considered promising adjuvant therapeutic agents alongside standard cytostatic drugs [9,20].

Similarly, *Piper nigrum* (black pepper) has been widely used for centuries, not only as a spice but also for its medicinal properties [15]. P and its derivatives, key bioactive compounds in *Piper nigrum* extracts, have shown anticancer effects in melanoma, breast, and colon cancer by modulating oxidative stress and apoptosis [15,21]. Unlike isolated compounds, whole *Piper nigrum* extracts may provide enhanced therapeutic benefits due to the combined action of multiple bioactive constituents, potentially leading to synergistic effects that increase their anticancer potential [22,23,24].

**Table 1 ijms-26-05688-t001:** Chemical structures of compounds found in *Cannabis sativa* flowers and black pepper fruits (*Piper nigrum*) were analyzed in this study. CBC, CBG, CBDA, P, CBD, THC, and BCP.

Compound	Chemical Structure	Effects	References
Cannabichromene (CBC)	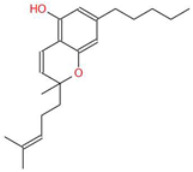	A non-psychoactive cannabinoid with anti-inflammatory and anti-cancer properties, potentially modulating tumor cell proliferation and apoptosis.	[3,10]
Cannabigerol (CBG)	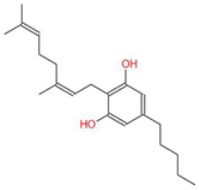	Known as the precursor for many cannabinoids, it demonstrates anti-proliferative effects on cancer cells and modulates the endocannabinoid system.	[5,12]
Cannabidiolic Acid (CBDA)	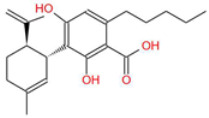	A precursor to CBD, this compound exhibits anti-nausea and possible antitumor properties.	[13,25]
Piperine (P)	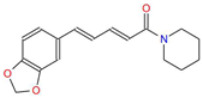	The major bioactive alkaloid in *Piper nigrum*, acts as a TRPV1 agonist, influencing apoptosis and oxidative stress in tumor cells.	[11,18]
Cannabidiol (CBD)	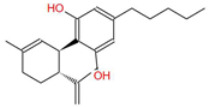	A widely studied cannabinoid with anti-inflammatory, anti-tumor, and neuroprotective properties.	[6,13]
Tetrahydrocannabinol (THC)	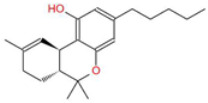	The principal psychoactive compound in *Cannabis sativa*, also known to induce apoptosis in cancer cells.	[14,15,26,27]
Beta-caryophyllene (BCP)	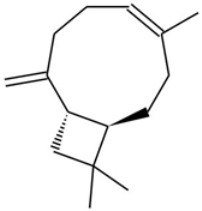	A sesquiterpene with anti-inflammatory and potential anti-cancer activities, acting as a CB2 receptor agonist.	[12,19]

Emerging evidence highlights molecular targets like GPR55 and PINK1 as key players in glioblastoma progression. GPR55, a non-canonical cannabinoid receptor, is implicated in cell proliferation, migration, and invasion [16,28]. Functionally, GPR55 contributes to cancer progression by activating ERK signaling, which promotes proliferation, and by inducing AKT phosphorylation, a key driver of survival pathways in ovarian carcinoma (OVCAR-3) and prostate cancer cells [29]. GPR55 overexpression is associated with poor prognosis in glioblastoma patients, correlating with lower overall survival [28,30]. Similarly, in advanced stages of pancreatic ductal adenocarcinoma, elevated GPR55 expression is linked to increased tumor aggressiveness [28]. In contrast, in human squamous cell carcinoma, GPR55 upregulation has been proposed as a novel biomarker for disease characterization [31]. Analyses of glioblastoma patients with high versus low GPR55 expression further support its prognostic relevance, as elevated GPR55 mRNA levels within the tumor are associated with reduced survival [28,30]. Likewise, in colorectal cancer, high tumor-specific GPR55 mRNA expression significantly correlates with decreased relapse-free survival [28,32].

PINK1, a mitochondrial kinase, plays a crucial role in regulating mitophagy and maintaining cellular homeostasis [17,33]. Targeting mitophagy-related genes such as PINK1 could enhance cancer cell sensitivity to treatment by disrupting mitochondrial clearance, leading to the accumulation of dysfunctional mitochondria, increased oxidative stress, and, ultimately, cell death [34,35]. The dysregulated expression of these proteins has been associated with poor prognosis in glioblastoma and other malignancies [18,19,20,21,36,37,38,39], highlighting their potential as valuable biomarkers and therapeutic targets.

This study integrates in silico, in vitro, and ex vivo approaches to evaluate the antitumor potential of CBC, CBG, and other cannabinoids, alongside *Piper nigrum* derivatives, against malignant glioma. Molecular docking simulations were performed using PINK1 and GPR55 structures obtained from the AlphaFold database, enabling high-accuracy predictions of ligand–receptor interactions. Ligands, including cannabinoids and P, were optimized using the MMFF94 force field and assessed via AutoDock and Vina to calculate binding affinities and identify potential therapeutic candidates.

In parallel, cell viability assays, mitochondrial potential analyses, and immunofluorescence studies were conducted in glioblastoma cell models to validate bioinformatic predictions and assess functional responses. Additionally, immunohistochemical (IHC) analyses of PINK1 and GPR55 were performed on glioblastoma samples from Colombian patients, providing insights into their expression patterns in tumor tissues. Furthermore, various online tools and software, such as SwissADME, admetSAR 2.0, and pkCSM, were used to predict the toxicity and permeability profiles of the compounds. Finally, data mining across multiple transcriptomic and expression profiling databases was employed to complement the experimental findings.

The findings of this research aim to provide knowledge for the development of novel therapies for malignant glioma and identify potential molecular targets for future therapeutic development.

## 2. Results

The docking results for the PINK1 are presented in Table 2. The results show that osimertinib (OSI), piperine (P), and THC exhibit the strongest binding affinities, with consistent ranking across the three scoring functions. We included osimertinib (OSI) in our study while excluding other chemotherapy agents because, as a tyrosine kinase inhibitor, it aligns with our objective of evaluating the potential modulation of PINK1 given that it is a kinase itself. OSI stands out by presenting the highest Vina affinity (−9.30 kcal/mol) and the top exponential consensus ranking (ECR). On the other hand, P shows the best Autodock affinity (−9.75 kcal/mol). THC also shows strong affinity (−8.70 kcal/mol) with respect to the Vina scoring. CBG and CBDA presented high Vinardo affinities and were moderate in other functions. Moreover, CBD, CBC, and BCP displayed moderate affinities values. Hence, OSI, P, THC and CBG demonstrated the best binding affinities for PINK1.

The docking results for GPR55 in Table 3 indicate that CBC, OSI, P, and CBG exhibit the strongest binding affinities. CBC ranks first in Vina (−9.20 kcal/mol) and second in Autodock (−9.40 kcal/mol), achieving the highest ECR of 0.46. OSI follows with consistently strong performance, ranking first in Autodock (−9.46 kcal/mol) and third in Vinardo (−8.60 kcal/mol), with an ECR of 0.41. P ranks second in Vina (−8.50 kcal/mol) and third in Autodock (−8.36 kcal/mol), sharing an ECR of 0.36 with CBG, which also demonstrates notable binding, ranking first in Vinardo (−8.80 kcal/mol). These compounds exhibit promising affinity for GPR55 and suggest a potential role in modulating the protein.

We analyzed how the ligands interact with the PINK1 protein, and the results showed that they bind in different ways (Figure 1). OSI binds strongly through hydrophobic interactions with residues like Ile162, Leu369, and Met318, and it also forms electrostatic interactions with Asp366, Asn367, and Lys164. P forms a hydrogen bond with Tyr321 and interacts with other residues like Val381 and Arg276. THC shows hydrogen bonding with Gln327, along with hydrophobic interactions with Val170 and Asp366. CBG and CBDA both form hydrogen bonds with Glu371, and they interact hydrophobically with Trp379 and Val184. CBD also binds via hydrogen bonds with Lys319, as well as hydrophobic interactions with Trp379 and Tyr321. BCP interacts mainly through hydrophobic forces with Ile162 and Leu369. The compounds interact with critical residues, including Asp362, Asn367, Lys364, Asp366, Tyr321, Cys548, and Glu371, which play essential roles in the catalytic activity and functional regulation of PINK1 [38,39,40,41,42,43]. Overall, these ligands interact with PINK1 through a mix of hydrophobic, electrostatic, and hydrogen bonding interactions, which could help guide the development of targeted therapies for PINK1.

For the interactions of the ligands with GPR55 (Figure 2), key hydrophobic interactions with residues like Phe102, Phe159, Phe188, and Leu148 are involved in the binding of several ligands, including CBC, CBG, CBD, THC, OSI, P, and BCP. Additionally, hydrogen bonds were observed for some ligands, and CBC formed one hydrogen bond with Val149; OSI with Gly189; THC with Leu185 and Val181; and CBDA with Arg253, Met172, Gln249, and Asn171. CisPt exhibited several hydrogen bonds with Asn280, Ser277, Cys281, Asn278, and Asp70. Other ligands like P, CBG, CBD, BCP, and TMZ interacted only through hydrophobic contacts, commonly involving residues like Ser153, Tyr106, Phe110, Gly152, Val149, and Leu148. The ligands interact with residues central to GPR55’s ligand-binding pocket, such as Phe102, Tyr106, Ser153, Gly152, and Val149 [42,43,44,45,46,47]. Compounds like OSI, CBC, P, and CBG show strong engagement with these residues, aligning with their high docking scores. Other compounds, such as CBDA and CisPt, primarily interact with extracellular or regulatory residues, hinting at the less direct modulation of the receptor’s active conformation. Overall, similarly to their interactions with PINK1, these ligands bind to GPR55 through diverse interactions, suggesting their potential to modulate the activity of both proteins.

After performing molecular docking simulations, we analyzed the compounds using the SwissADME, admetSAR 2.0, and pkCSM databases. First, we generated the “boiled egg” diagram (Figure 3), which presents a graphical prediction of passive intestinal absorption and brain penetration as a function of lipophilicity and apparent polarity (described by WLOGP and TPSA, respectively). The diagram also includes the color coding of the data points to indicate the predicted active efflux by the P-glycoprotein, as explained in the legend on the right side of the plot. Most analyzed compounds, except for TMZ and CisPt, exhibited acceptable human intestinal absorption (HIA). Regarding blood–brain barrier (BBB) permeability, the compounds with the lowest predicted probability of crossing the BBB were OSI, TMZ, and CisPt. This finding aligns with the known pharmacokinetic properties of these chemotherapeutic agents used in gliomas. Notably, OSI was the only compound identified as P-glycoprotein-positive (P-gp+), which is consistent with the literature. Tyrosine kinase inhibitors (TKIs) [48], such as OSI, are known to be substrates of ATP-binding cassette (ABC) transporters, which can impact their pharmacokinetics and contribute to drug resistance in cancer patients. Additionally, TKIs have been reported to inhibit P-gp ATPase activity in a dose-dependent manner, further influencing their therapeutic efficacy [48].

Secondly, we generated Table 4, which outlines the in silico toxicity profiles of the analyzed compounds.

The toxicity analysis of these compounds shows that most cannabinoids do not exhibit significant toxicity in the tests conducted. None of them display mutagenicity (AMES test), hERG I inhibition, or carcinogenicity, indicating a favorable safety profile in these aspects. However, P and the drug OSI exhibit hepatotoxicity, suggesting a potential risk to the liver. Additionally, BCP and CBG show skin sensitization, implying a potential for allergic skin reactions. In contrast, TMZ presents mutagenicity, carcinogenicity, and hepatotoxicity, reflecting its nature as a chemotherapeutic agent with significant adverse effects.

In parallel with the bioinformatic analysis, immunohistochemical (IHC) samples were collected and analyzed, with the results summarized in Figure 4. The expression of PINK1, GPR55, and mutant p53 was evaluated in glioblastoma patient samples. Among the 20 cases analyzed, PINK1 was expressed in 65% of the samples (13 positive and 7 negative), GPR55 exhibited high expression in 95% (19 positive and 1 negative), and p53 mutations were present in 30% (6 positive and 14 negative). These findings reveal distinct expression patterns that may correlate with glioblastoma characteristics. The detailed anatomical and molecular features of the selected patients are provided in Appendix A, which outlines the clinical profiles of the Colombian cohort with available tumor samples for IHC assays. Additionally, the brain-tissue-processing procedure is illustrated in the Appendix A.

A key observation from the IHC analysis was that PINK1 expression was detected in neurons and endothelial cells but absent in normal glial cells. However, in glioblastoma samples, PINK1 was present in positive cases, suggesting a tumor-specific role. In contrast, GPR55 was expressed in both control and tumor tissues, with significantly higher intensity in glioblastoma samples, indicating a potential role in tumor progression.

This figure compares the expression of PINK1 and GPR55 in glioblastoma tumor tissues versus a non-tumoral human cortex, highlighting their potential associations with glioblastoma characteristics. Immunohistochemical analysis revealed that PINK1 was expressed in 65% of tumor samples (13 positive and 7 negative), while GPR55 exhibited high expression in 95% of cases (19 positive and 1 negative). Additionally, p53 mutations were detected in 30% of the samples (6 positive and 14 negative). These findings suggest that PINK1 and GPR55 are highly expressed in glioblastoma tissues, potentially implicating them in the disease’s pathophysiology (10× magnification and 100 µm scale bar).

Subsequently, a series of cell viability assays were performed using the U87MG, T98G, CCF-STTG1, SH-SY5Y, and MO3.13 cell lines (Figure 5, Figure 6 and Figure 7 and Appendix A). Table 5 summarizes the IC_50_ values for various compounds analyzed across different cell lines, presented in both ng/μL and μM. The following compounds were tested: CBC, CBG, CBD, CBDA, ethanol extract of *Piper nigrum* (PiperOH), essential oil of *Piper nigrum* (PiperEO), piperine (P), cisplatin (CisPt), temozolomide (TMZ), and osimertinib (OSI).

Table 5 presents the IC_50_ values of selected compounds in various cancer cell lines expressed in ng/μL and μM. Among astrocytoma cell lines, CBC, CBG, and CBD exhibited notable cytotoxic activity, with IC_50_ values ranging from 1.56 to 6.25 ng/μL in U87MG, T98G, and CCF-STTG1 cells. In contrast, CBDA showed lower potency, particularly in U87MG (9.3 ng/μL) and SH-SY5Y (9.4 ng/μL, ~29.9 μM) under CBD treatment. The *Piper nigrum* derivatives displayed variable activity, with PiperOH, PiperEO and P being non-cytotoxic (NC) in some models. The mean IC_50_ values across astrocytoma models for CBC and CBG were 9.88 ± 2.8 μM and 13.24 ± 4.3 μM, respectively, highlighting their potential as therapeutic candidates.

A further analysis was conducted on the CCF-STTG1 cell line, as detailed in Figure 5, Figure 6 and Figure 7.

Due to the effects observed with PiperOH extracts, a further investigation was conducted to evaluate the impact of combining these extracts with cannabinoid compounds and their fractions. The goal was to determine whether the cytotoxic effect was altered or completely inhibited. Notably, in the CCF-STTG1 cell line, as shown in Figure 5, the presence of PiperOH, PiperEO (which contains at least 30% β-caryophyllene), and P with 95% purity increased cell survival. Additionally, Figure 7 illustrates the combination of PiperOH and P with CBD and CBG, revealing both the mitigation and inhibition of cytotoxicity, as well as morphological changes in CCF-STTG1 cells treated with these compounds. Compared to the healthy control, CCF-STTG1 cells exposed to high concentrations (25 ng/μL) maintained their viability and cellular integrity when treated with PiperOH, whereas signs of cell death were evident when exposed to pure cannabinoids alone. These findings suggest that piperine-derived compounds may modulate the cytotoxic effects of cannabinoids, highlighting the need for further mechanistic studies.

Finally, but no less importantly, these experiments were also conducted in the T98G and U87MG cell lines. However, the results were not significant, as the combination did not reduce the cytotoxicity of cannabinoids in these models. This suggests that the observed protective effect of PiperOH, PiperEO, and P may be specific to certain glioma cells, such as CCF-STTG1, rather than a generalizable phenomenon across all tested models.

In addition to viability assays, mitochondrial membrane potential was assessed using an Immuno Tracker in the astrocytoma cell lines listed in Table 5. Changes in mitochondrial potential correlating with cell viability across different treatment conditions were observed in Figure 8. Notably, cannabinoid compounds in the enumerated models tended to decrease the membrane potential, which could be associated with the cytotoxic effects of these compounds, dependent on mitochondrial integrity, which is, among other factors, regulated by PINK1. Furthermore, Appendix A highlights the changes in PINK1 and GPR55 expression in some of the astrocytoma cell lines analyzed throughout the study with immunofluorescence (IF) under various treatments. The reduction in the intensity of expression of these proteins was observed as a result of the treatments shown, correlated with the cytotoxicity exhibited.

Finally, with all the in silico and in vitro data obtained regarding the expression of PINK1 and GPR55 and their potential modulation in cancer models, we decided to compare these findings with data from public databases. Data collection and analysis were performed using the UALCAN, GEPIA, GLIOVIS, and Human Protein Atlas databases, as shown in Figure 9, Figure 10 and Figure 11, Table 6, and the Appendix A. In Figure 9, generated with GLIOVIS, Kaplan–Meier survival curves were plotted for glioblastoma subtypes correlated with PINK1 expression. In Figure 10, Kaplan–Meier curves were generated based on PINK1 expression levels in other cancer types (lung, pancreatic, and renal) using the Human Protein Atlas database. Additionally, the Kaplan–Meier curves related to GPR55 and PINK1 expression data in GBM vs. low-grade glioma (LGG) are shown in Appendix A, which includes information from GLIOVIS and GEPIA. Finally, the boxplots in Figure 11, derived from the GEPIA1 database, illustrate significant variations in PINK1 expression across different tumor types, highlighting its potential to influence survival outcomes based on the molecular and cellular context.

## 3. Discussion

There is an unmet need to find new therapeutic approaches that can improve the prognosis and survival of patients suffering from malignant gliomas, especially GBM. Among the potential therapeutic targets for these conditions are the mitochondrial protein PINK1 and GPR55 receptors, as the modulation of these has been shown to affect the progression of these cancers in various models [48,49,50]. Additionally, in recent years, ECS has emerged as a major area of interest in multiple pathologies, including cancer, due to its ubiquitous distribution and the findings of ECS-related disorders in cancer [51,52].

In this context, the molecular docking study (Table 2 and Table 3 and Figure 1 and Figure 2) provides valuable insights into the binding affinities and interactions of selected compounds with the PINK1 and GPR55 proteins. Key compounds, such as CBC, P, and THC, exhibited strong and consistent binding to these targets, highlighting their potential as therapeutic modulators of PINK1 and GPR55, both of which are involved in various cellular and pathological processes.

For PINK1, it is worth noting that OSI demonstrated the highest binding affinity, particularly with the Vina scoring function (−9.30 kcal/mol). Its interaction profile includes strong hydrophobic contacts with residues such as Ile162, Leu369, and Met318, along with crucial electrostatic interactions with Asp366 and Lys164. These interactions likely stabilize the OSI-PINK1 complex, reinforcing its potential as a potent modulator of PINK1 activity. OSI, also known as AZD9291, is a third-generation epidermal growth factor receptor tyrosine kinase inhibitor (EGFR-TKI) approved for the treatment of non-small cell lung cancer (NSCLC) with a T790M mutation in EGFR. It demonstrated significant efficacy against various cancers in both preclinical and clinical trials [53]. Since EGFR is one of the most frequently mutated genes in GBM, we evaluated its potential as a therapeutic target.

Additionally, P demonstrated strong binding affinities, achieving the highest Autodock score (−9.75 kcal/mol). Its interaction with Tyr321, through hydrogen bonding, along with hydrophobic contacts involving Val381 and Arg276, underscores its high binding potential. Similarly, THC (−7.35 kcal/mol) exhibited a well-balanced interaction profile, forming hydrogen bonds and hydrophobic interactions with residues such as Gln327 and Val170.

Compounds like CBG and CBD demonstrated moderate binding affinities, primarily interacting with residues associated with PINK1’s catalytic activity. In contrast, TMZ and CisPt displayed weaker binding affinities, suggesting the direct limited modulation of PINK1 by these agents, consistent with the canonical mechanism of action of alkylating agents.

For GPR55, CBC and P emerged as top-performing ligands. CBC exhibited the strongest binding affinity with Vina (−9.20 kcal/mol) and ranked highly across other scoring functions. Its key interactions include hydrophobic contacts with residues such as Phe102 and Leu148, along with a hydrogen bond with Val149, suggesting its potential as a potent GPR55 modulator. OSI also demonstrated high binding affinities, forming hydrogen bonds with Gly189 and engaging in hydrophobic interactions with residues like Phe159 and Leu148. Similarly, P and CBG consistently interacted via hydrophobic contacts across the GPR55 ligand-binding pocket. In contrast, CisPt, despite forming hydrogen bonds with Asn280 and Cys281, exhibited weaker binding affinities, limiting its potential as a GPR55 modulator. Additionally, compounds like BCP and TMZ displayed minimal direct interactions with the receptor’s active site, indicating a lower modulatory potential.

The docking results reveal distinct interaction profiles between PINK1 and GPR55 for the tested compounds. CBC and P emerge as strong modulators of GPR55, and THC and CBDA display better affinities for PINK1. This differential affinity could guide the design of selective or dual-targeted therapeutic agents based on disease-specific requirements.

Interestingly, in silico analyses suggest that P and CBC are promising candidates for modulating pathways involving PINK1 and GPR55. Their interactions with key residues, through both hydrophobic and electrostatic forces, emphasize the importance of optimizing ligand structures in enhancing these interactions. The diversity in binding modes further suggests potential for tailoring compounds to achieve specificity for one target over the other.

While these docking simulations provide significant insights, the absence of crystallized structures for PINK1 and GPR55 introduces uncertainties. Although AlphaFold-generated models are reliable, they remain predictive rather than experimentally validated. To strengthen these findings, future studies should incorporate experimental approaches such as site-directed mutagenesis, X-ray crystallography, or cryo-electron microscopy.

We performed additional analysis of BBB permeation and toxicity in selected compounds shown in Figure 3 and Table 4. The boiled egg diagram (Figure 3) provides a detailed analysis of the pharmacokinetic properties of the compounds based on their WLOGP (lipophilicity) and TPSA (topological polar surface area). Compounds **1**, **2**, **3**, **5**, **6**, and **7** are located within the yellow (yolk) region, indicating a high probability of penetrating the brain via the BBB while also being in the white region, which suggests good gastrointestinal absorption. Compound **4** is in the white region but outside the yellow region, indicating gastrointestinal absorption but reduced brain penetration. Compound **8** (OSI, in blue) is a PGP+ substrate, meaning that active efflux by the P-glycoprotein limits its brain penetration despite being in the white region. Compounds **9** (TMZ) and **10** (CisPt), while positioned outside the yellow region and showing limited potential for BBB penetration due to high TPSA values, are widely used as antitumor agents in the clinical market. Notably, compounds **9** and **10** are extensively employed in glioblastoma treatment despite their pharmacokinetic limitations regarding BBB penetration, and this is likely due to their efficacy in targeting tumor cells and their established therapeutic profiles. This analysis highlights compounds **1**–**7** as having favorable absorption profiles, with some also showing brain penetration potential, while compounds **8**, **9**, and **10** underscore the importance of considering both pharmacokinetics and therapeutic context when evaluating drug efficacy.

Table 4 presents the in silico toxicity profiles of various compounds analyzed, including cannabinoids (CBC, CBG, CBD, CBDA, and THC), P, BCP, and reference drugs OSI and TMZ. Notably, none of the compounds exhibited AMES toxicity or hERG channel inhibition, indicating low mutagenic and cardiotoxic potential. However, hepatotoxicity was identified in P, OSI, and TMZ, highlighting a potential risk for liver damage with these compounds. Skin sensitization was observed exclusively with CBG and P, suggesting a selective allergenic response. Importantly, carcinogenicity was absent across all compounds, except for TMZ, which is a known chemotherapeutic agent with established carcinogenic risks. These findings underscore the relatively safe toxicity profiles of cannabinoids compared to synthetic drugs, although specific risks, such as hepatotoxicity and skin sensitization, warrant further investigation for some compounds.

On the other hand, Figure 4, Appendix A provide insights into the immunohistochemical (IHC) analysis conducted in parallel with the bioinformatic study. The expression patterns of PINK1, GPR55, and mutant p53 were evaluated in glioblastoma patient samples, revealing that PINK1 was detected in 65% of cases (13 positive and 7 negative), GPR55 was highly expressed in 95% (19 positive and 1 negative), and p53 mutations were present in 30% (6 positive and 14 negative). Notably, the clinical data from the Colombian patient cohort align with global epidemiological trends, showing a higher prevalence of glioblastoma in men compared to women and a peak incidence in individuals aged 50–60 years. The presence of p53 mutations in 30% of cases is also consistent with worldwide statistics, reinforcing the relevance of this cohort in glioblastoma research. Further details on patient demographics and molecular profiles are summarized in Appendix A, while Appendix A illustrates the tissue-processing methodology. These findings highlight the potential implications of PINK1 and GPR55 in glioblastoma and support their relevance as biomarkers in the disease’s pathophysiology.

Figure 5, Figure 6 and Figure 7 provide evidence of the cytotoxic potential of cannabinoids and *Piper nigrum* bioactives relative to glioblastoma cell lines, including MO3.13 (oligodendrocyte precursor cells) and SH-SY5Y (neuroblastoma). The results indicate that in almost all cell lines (U87MG, T98G, MO3.13, and SH-SY5Y), both *Piper nigrum* derivatives and cannabis compounds exhibit dose-dependent cytotoxicity, reducing cell viability at higher concentrations. Table 5 summarizes the IC_50_ values obtained in this study, with some data similar to those reported by other laboratories. Tamara Lah and collaborators, who evaluated CBG and CBD in glioblastoma, demonstrated that CBG exhibit significant effects on reducing cell viability in glioblastoma cell lines, with IC_50_ values at 28.1 ± 1.1 μM, similarly to THC (IC_50_ 27.9 ± 1.8 μM). CBD, however, showed greater cytotoxicity with an IC_50_ of 22.0 ± 2.1 μM, which was significantly more effective than CBG and THC [12,13]. The analysis of our study reveals significant variability in the cytotoxicity of the compounds, with some showing higher potency across specific cell lines. CBC and CBG demonstrated notable activity in the T98G and CCF-STGG1 cell lines, with IC_50_ values ranging from 1.56 ng/μL to 3,12 ng/μL (4.93 μM to 9.91 μM). PiperOH exhibited cytotoxicity, particularly in the U87MG, SH-SY5Y and T98G cell lines, showing effective results as low as 6.25 ng/μL. CBDA, in comparison to CBD, showed moderate activity in most cell lines with less pronounced effects than CBG and CBC. When comparing CBDA, CBG, and CBC with TMZ (200–250 μM [54,55]) and CisPt (>125 μM [56]), for cancer therapy, the potential of these compounds, especially in glioblastoma and other chemotherapy-resistant tumors, becomes evident due to their effectiveness at lower doses compared to the established standards. In contrast, compounds like PiperOH and PiperEO displayed non-cytotoxic effects (NC*) in some cell lines, indicating their limited or absent anticancer activity in these settings.

Figure 6 shows the potential additive effect of CisPt with CBG compared to CBD in the astrocytoma cell line CCFSTTG1. Preliminary results suggest that CBG may enhance the therapeutic efficacy of drugs, potentially expanding treatment options in oncology. This is particularly relevant for aggressive cancers such as glioblastoma, where therapeutic resistance and poor prognosis remain major challenges.

Furthermore, consistent with our findings, data from the Human Protein Atlas suggest a potential correlation between PINK1 and GPR55 expression profiles and the treatment responses observed in this study (Appendix A). Specifically, PINK1 expression follows the order U87MG > T98G > CCF-STTG1 > SH-SY5Y, while data for the MO3.13 cell line remain inconclusive. Regarding GPR55, Appendix A shows negligible expression in the cell lines evaluated in this study. However, previous reports indicate considerable GPR55 expression in SH-SY5Y [55,56], T98G [57], and U87MG [58] cells and other models [59,60], suggesting its potential role in glioblastoma pathophysiology and therapeutic responses.

Moreover, the immunofluorescence analysis (IF) (Appendix A) of PINK1 and GPR55 in U87MG and T98G cells revealed that treatment with cannabinoids such as CBG and CBD reduced fluorescence intensity, indicating lower PINK1 expression. This effect correlates with mitochondrial membrane potential assessments (Figure 8), where a decrease was observed in astrocytoma cell lines following treatment with these cannabinoids, which aligns with the cytotoxicity results obtained via MTT assays. Notably, TMZ and P had differential effects on mitochondrial potential depending on the astrocytoma subtype. It is worth mentioning that, although CBC induces cell death, it does not appear to alter the intensity of PINK1 or GPR55 expression in U87MG and T98G cell lines, suggesting that its cytotoxic effect may be mediated through a different mechanism. Additionally, immunofluorescence analysis (Appendix A) revealed that the cannabinoid receptor type 1 (CB1) is widely expressed in U87MG glioblastoma cells. Interestingly, while cannabinoids such as CBG also induce cell death, they do not reduce CB1 expression, indicating that their effects on glioblastoma cells may involve alternative or complementary pathways beyond CB1 modulation.

While these results suggest a possible trend, further studies are needed to confirm their clinical significance. To our knowledge, this is the first published IHC report on GPR55 and PINK1 expression in glioblastoma patient samples, providing novel insights into their potential relevance in glioblastoma pathophysiology.

To further investigate whether the *Piper nigrum* extract modulates the cytotoxic profile of cannabinoids (CBD and CBG) when used concomitantly, we focused on the CCF-STTG1 cell line (Figure 6). Interestingly, *Piper nigrum* did not exhibit cytotoxic effects in the CCF-STTG1 cells; instead, both PiperEO and PiperOH increased cell viability. Moreover, the concomitant use of cannabinoids with *Piper nigrum* derivatives resulted in a complete or apparent blockade of cannabinoid-induced cytotoxicity in CCF-STTG1 cells (Figure 7a,b) in contrast to the T98G and U87MG models. This suggests a distinct interaction between these compounds in this specific cellular context.

To determine whether purified P from *Piper nigrum* was responsible for this antagonistic effect, we assessed its impact on cell viability. While P alone had no effect or even increased cell viability at high doses, previous studies have reported its cytotoxicity in various cancer models [61,62]. However, when used concomitantly with cannabinoids, F did not appear to alter the cytotoxic response, suggesting that another component of the *Piper nigrum* fraction may be responsible for this effect, warranting further investigation.

Figure 7b further explores the cellular effects of these treatments, highlighting morphological changes indicative of apoptosis. Cells treated with CBD or CBG exhibited classic apoptotic features, such as membrane blebbing and cellular shrinkage. Notably, the addition of *Piper nigrum* extracts to CBG or CBD treatment reduced these apoptotic effects, reinforcing the observed modulation of glioma cell responses by these compounds.

The CCF-STTG1 cell line, classified as a grade IV mixed astrocytoma according to the WHO 2021 classification of CNS tumors, presents a unique context for interpreting these results [63]. One of the main findings lies in the pro-viability effects of *Piper nigrum* derivatives observed in this cell line, contrasted with their cytotoxic effects in other models. In the CCF-STTG1 cell model, *Piper nigrum* derivatives do not appear to exert direct cytotoxic effects and instead attenuate the activity of cannabinoids. One possible explanation is that these compounds mitigate oxidative stress and mitochondrial damage induced by cannabinoids before the activation of cell death. Interestingly, Figure 8C shows that treatment with PiperOH increases mitochondrial potential, possibly as a protective mechanism to prevent mitochondrial depolarization and safeguard the cell from apoptosis. This suggests that PiperOH may play a role in preserving mitochondrial function under stress conditions induced by cannabinoids, and this specific cell subtype could have a distinct vulnerability that *Piper nigrum* could modulate, potentially opening new avenues for targeted therapeutic approaches.

Another potential mechanism involves the interference of *Piper nigrum* derivatives with ABC transporter family members [64,65]. Which could affect the bioavailability and intracellular transport of cannabinoids in these cells.

Additionally, the apparent attenuation or blockade of the cytotoxic effects of cannabinoids in the CCF-STTG1 model suggests a potential mechanistic overlap. Based on in vitro results and molecular docking analyses, a plausible explanation is that *Piper nigrum* derivatives modulate PINK1 function by competing for its binding, thereby altering its role in mitochondrial homeostasis and cellular phosphoproteome regulation. This competitive interaction could mitigate the pro-apoptotic effects typically induced by cannabinoids, particularly in a tumor context with complex metabolic dynamics, such as that of the CCF-STTG1 cell line.

To further elucidate this phenomenon, complementary biochemical studies, such as Western blotting (WB) and PCR assays, are currently underway within our research group. These investigations aim to clarify the interaction between PINK1 and GPR55 modulation, mitochondrial dynamics, and the observed effects on cell viability. Understanding these mechanisms could pave the way for more precise and effective therapeutic strategies, particularly for glioblastoma and other CNS tumors.

Building on the molecular interaction data from docking studies and the in vitro viability results discussed earlier, additional insights were gained through data mining across multiple transcriptomic and expression profiling databases, including UALCAN, GLIOVIS, TCGA, GEPIA, and the Human Protein Atlas. These platforms revealed patterns mainly of PINK1 expression in glioblastoma tissues compared to normal brain samples. In glioblastoma datasets from UALCAN and TCGA, PINK1 expression was consistently downregulated, aligning with the hypermethylation observed in Appendix A. This suggests that PINK1 is epigenetically silenced in glioblastoma, potentially impairing mitochondrial quality control and contributing to tumor progression. The comparison between normal and primary tumor tissue samples shows an extremely significant difference in PINK1 expression, with a *p*-value of 1.11 × 10^−16^. This suggests that PINK1 expression is significantly lower in primary tumors (*n* = 156) compared to normal tissues (*n* = 5). This drastic reduction in expression may indicate a crucial role of PINK1 in glioblastoma biology (as a potential indicator of tumor progression). This aligns with previous findings that PINK1 supports cellular survival by maintaining mitochondrial quality control. However, its downregulation in glioblastoma may reflect an adaptive response to the tumor microenvironment’s metabolic demands, potentially favoring tumorigenesis. These findings highlight PINK1 as a promising biomarker and therapeutic target, particularly for strategies aimed at reversing epigenetic silencing. GPR55 data were not included, as no significant results were observed.

Interestingly, the Kaplan–Meier survival analysis in Figure 9 using Gliovis reveals a subtype-dependent relationship between PINK1 expression and patient prognosis. In proneural glioblastoma subtypes, high PINK1 expression correlates with poorer survival, suggesting a pro-oncogenic role in specific molecular contexts. Conversely, in mesenchymal and classical subtypes, no significant association is observed, highlighting the complexity of PINK1’s role in glioblastoma biology. This context-dependent variability underscores the need for stratifying glioblastoma patients based on molecular profiles when considering PINK1-targeted therapies.

Figure 10 and Figure 11 expand the analysis of PINK1’s role in cancer, emphasizing its tumor-specific variability and prognostic potential. The Kaplan–Meier survival analysis in Figure 9 reveals that PINK1 expression levels correlate with patient outcomes in a tumor-type-dependent manner. In malignancies such as kidney renal clear cell carcinoma (KIRC), high PINK1 expression is associated with better survival rates, indicating a protective role. Conversely, in lung squamous cell carcinoma (LUSC) and PAAD, elevated PINK1 expression corresponds to poorer survival outcomes, suggesting a pro-oncogenic function in these contexts. These findings underscore PINK1’s dual role in cancer biology, acting as either a tumor suppressor or promoter depending on the tumor microenvironment and molecular landscape.

It is worth mentioning that some studies point to PINK1 as a potential biomarker [66] and therapeutic agent. For example, some studies demonstrate that PINK1 depletion in the non-small cell lung cancer (NSCLC) A549 cell line via shRNA reduced cancer cell proliferation, increased cell death, decreased ATP production, inhibited mitophagy, and increased ROS levels and caspase-9-dependent apoptosis [67]. These findings suggest that PINK1 depletion disrupts energy metabolism and increases sensitivity to glycolysis inhibitors. In this context, targeting the accelerated metabolic activity of tumor cells could be a promising therapeutic strategy while minimizing damage to non-malignant tissues. Furthermore, in breast cancer, PINK1 expression has been significantly correlated with histological grade, and its depletion induces a distinct proteomic profile, particularly impacting energy metabolism pathways. These results support an oncogenic role for PINK1 in breast cancer, offering new insights into its potential regulatory function [68].

The boxplot in Figure 11 further highlights the significant differences in PINK1 expression between tumor and normal tissues across various cancer types, including GBM, LGG, KIRP, and others. In GBM and PAAD, PINK1 expression is notably downregulated in tumor samples, aligning with its reported epigenetic silencing in these malignancies. In contrast, its upregulation in renal cancers, such as KIRC and KIRP, suggests that PINK1 may have context-dependent protective effects.

These data reflect PINK1’s potential to function as a molecular switch, influenced by the tumor’s genetic background, metabolic demands, and microenvironmental factors. For instance, in glioblastoma, the loss or gain of PINK1 may contribute to mitochondrial dysfunction and altered mitophagy, affecting tumor progression. Meanwhile, in renal cancers, its upregulation might enhance mitochondrial integrity, supporting cellular resistance against metabolic stress.

The prognostic variability of PINK1 demonstrated in Figure 10 and Figure 11 underscores the importance of integrating molecular profiling into cancer treatment strategies. Beyond glioblastoma, these insights reveal opportunities to explore PINK1 as a therapeutic target or biomarker in a broader range of cancers. However, its dual role necessitates further investigation to delineate the mechanisms underlying its protective versus pro-oncogenic functions.

It is worth mentioning that, although no significant correlation between GPR55 and GBM was found in the databases consulted for this study, when comparing the expression data and Kaplan–Meier survival curves between LGG and GBM, significant differences in both GPR55 and PINK1 expression were observed, as shown in the Appendix A. These data were obtained from both the GLIOVIS and GEPIA databases, suggesting a potential role for these proteins in the natural course or progression of the disease. Additionally, while the role of PINK1 in cancer remains controversial [66,69], its possible dual function as a kinase dependent on the tumor subtype is consistent with the existing literature [66,70]. Regarding GPR55, it has been increasingly recognized as a potential oncogene in the carcinogenic development of GBM, as well as in other types of cancer [13,25,28,60].

Additionally, changes in the expression profiles of PINK1 and GPR55 were observed in correlation with survival rates in high-grade vs. low-grade gliomas, suggesting potential alterations in inflammatory and metabolic profiles [39,66,67,68,69,70,71,72]. These changes may correlate with malignancy and the natural progression of the disease, warranting further investigation to unravel this complex interplay.

As a final aspect to mention regarding PINK1, while RNA expression data (nTPM) indicate low levels of PINK1 (Appendix A), this does not necessarily mean that the protein is absent or not expressed. Our IHC results (Figure 4) revealed that PINK1 was present in neurons and endothelial cells but absent in normal glia, whereas in glioblastoma samples, positive cases (65%) exhibited PINK1 expression. This suggests a possible role in tumor progression, potentially through mitochondrial dynamics, metabolic adaptation, or resistance to apoptosis. Additionally, Kaplan–Meier survival analysis in glioma indicates that lower PINK1 levels correlate with better overall survival (Appendix A and Figure 9a), further supporting its potential oncogenic role in glioblastoma and highlighting the need for functional validation.

Furthermore, mitochondrial potential assays and immunofluorescence studies showed that compounds such as CBD and CBG led to mitochondrial membrane potential loss and reduced PINK1 expression, correlating with cytotoxic effects in glioblastoma cell lines. These findings indicate the direct and indirect modulation of PINK1, reinforcing the need for further investigation into its regulatory mechanisms in glioblastoma.

Beyond its role as a mitochondrial sensor, PINK1 has additional functions, including calcium [73] and iron homeostasis [74], phosphoproteome regulation [75], and potential immune modulation [76], reflecting its dualistic kinase activity. These diverse roles suggest that PINK1’s oncogenic potential in glioblastoma could be linked to broader cellular signaling pathways beyond mitophagy, emphasizing the need for further investigations to determine its therapeutic relevance in glioblastoma and other cancers.

Finally, it is particularly interesting and valuable to explore the therapeutic implications of cannabinoids in cancers like pancreatic cancer and glioblastoma, given their resistance to conventional treatments and the limited survival rates associated with these cancers. This highlights the urgent need to identify potential vulnerabilities for targeted therapeutic exploitation. To the best of our knowledge, this is the first study to test cannabinoids such as CBC and CBG in various glioma cell lines, correlating them with molecular docking studies and expression data from the IHC of patients and public human databases, specifically focusing on PINK1 and GPR55 expression.

Future studies should aim to further scale these investigations to better understand the underlying molecular pathways. Such work could help determine whether cannabinoids are a viable therapeutic option in oncology. Several preclinical studies already support the potential of plant-derived therapies in gliomas [77,78], and given the current challenges in treating these cancers, there is a pressing need to explore alternative therapies to improve patient outcomes and survival.

Prospective randomized clinical trials should also be conducted to evaluate the use of plant extracts as adjunctive agents in standard treatments. Recent clinical trials such as NCT05753007, NCT05629702, and NCT03529448 underscore the importance of plant-based research in glioblastoma treatment. On the other hand, TTX101, a hydrogel containing piperlongumine from *Piper longum*, is being tested for post-surgical application to target residual GBM cells due to its pro-apoptotic properties. Meanwhile, clinical trials investigating CBD and THC-based therapies aim to modulate immune responses and reduce inflammation, offering promising alternatives for improving glioma treatment.

## 4. Materials and Methods

### 4.1. Compounds

Cannabigerol (CBG, CAS Number: 25654-31-3, purity 99%), cannabidiol (CBD, CAS Number: 13956-29-1, purity 99%), broad-spectrum CBD resin (CBD > 80%, THC < 1%), cannabichromene (CBC, CAS Number: 20675-51-8, purity > 50%), and non-psychoactive cannabinoid fractions were obtained from Lasanta SAS, a Colombian medicinal Cannabis company located in Bogotá, Cundinamarca, Colombia. These compounds were sourced from *Cannabis* plants registered with the Colombian Agricultural Institute (Instituto Colombiano Agropecuario, ICA, Bogotá, Cundinamarca, Colombia, Resolution No. 00010394 of 2019, Figure 12a).

Additional compounds, including cannabidiolic acid (CBDA, CAS Number: 1244-58-2, purity 40%) and tetrahydrocannabinol (THC, CAS Number: 1972-08-3, purity 50%), were purified from extracts previously acquired by the Phytochemistry and Pharmacognosy Group (GIFFUN) at the Universidad Nacional de Colombia. An ethanolic extract and essential oil of *Piper nigrum* were prepared by GIFFUN using commercially sourced fruits cataloged in the Medellin Herbarium (AIP 9171 JAUM-101230, Figure 12b); further details are provided in the Appendix A. Piperine (CAS Number: 94-62-2, purity > 95%) was obtained from Abcam, Cambridge, MA, USA (Product Code: AB142933-1G).

Additional agents including cisplatin (CAS Number: 15663-27-1) was kindly donated by Josefa Rodriguez; temozolomide (TMZ, CAS Number: 85622-93-1) and Osimertinib (CAS Number: 1421373-65-0) were part of the inventory of the Universidad Nacional de Colombia and included for comparative analysis.

### 4.2. Sample Preparations

Stock solutions were prepared by dissolving 5 mg of each compound in 1 mL of filtered DMSO (0.22 μm pore size), yielding a final concentration of 5 mg/mL. A 5 μL aliquot from each stock solution was diluted into 1 mL of culture medium (RPMI or DMEM) supplemented with 2% fetal bovine serum (FBS), achieving a working concentration of 25 μg/mL. Serial dilutions were performed to construct dose–response curves for each compound.

For the cannabinoid and *Piper nigrum* fractions, the appropriate mass of each fraction was weighed, and the volumes corresponding to their purity were calculated to ensure the correct concentration. These concentrations were adjusted according to the purity of the fractions to maintain equivalence in final concentrations for the assays.

### 4.3. Tissue Samples and Cell Lines

The cell lines utilized in this study included U87MG, MO3.13, and SH-SY5Y (provided by the Universidad Nacional de Colombia); T98G (obtained from Universidad del Rosario); and CCf-STTG1 (donated by Universidad de los Andes). All cell lines were maintained in Dulbecco’s Modified Eagle Medium (DMEM) or RPMI medium, supplemented with 10% fetal bovine serum (FBS), 1% L-glutamine, and 1% penicillin/streptomycin. Cells were seeded in T-75 and T-25 flasks and incubated at 37 °C in a humidified atmosphere with 5% CO_2_.

The collection of glioblastoma patient samples was carried out using data from a database of 6010 cases provided by Bio-Molecular Diagnostica Ltd located in Bogota, Colombia, a specialized pathology laboratory, spanning the period from 2014 to 2024. From this dataset, 889 glioblastoma cases were selected and standardized according to the 2021 WHO classification of brain tumors, with a distinction made between IDH wild-type (IDH wt) statuses. After reviewing the inventory, 67 cases from 2024 were identified, and 20 cases with paraffin blocks were verified for use. A total of 40 slides were prepared in duplicate to evaluate the expression of PINK1 and GPR55. Non-pathological cortical brain tissue, which served as a control, was obtained from the Institute of Genetics, Universidad Nacional de Colombia.

### 4.4. Immunohistochemistry

Immunohistochemical analyses were performed to evaluate the expression of PINK1 (PINK1 antibody (38CT20.8.5): sc-517353; Santa Cruz Biotechnology, Dallas, TX, USA) and GPR55 (Anti-GPCR GPR55 antibody: ab174700 Abcam) antibodies in the glioblastoma samples. Tissue sections were deparaffinized in xylene and rehydrated through graded ethanol washes. Antigen retrieval was carried out using heat-induced epitope retrieval (HIER) in a citrate buffer (pH 6.0). Endogenous peroxidase activity was blocked using 3% hydrogen peroxide, followed by incubation with a protein block to prevent non-specific binding. Slides were incubated overnight at 4 °C with primary antibodies against PINK1 (dilution: 1:200) and GPR55 (dilution: 1:100). After rinsing, sections were treated with an HRP-conjugated secondary antibody, and staining was developed using a DAB substrate. Counterstaining was performed with hematoxylin, followed by dehydration and mounting with a permanent medium. The slides were examined under a light microscope to compare PINK1 and GPR55 expression in glioblastoma and control tissue samples, which was marked as absent or present [79].

### 4.5. Bioinformatic Analysis and Software Used

#### 4.5.1. Molecular Docking

The structures of the compounds were obtained from the PubChem database. The structures were optimized using the molecular editor Avogadro [80] and the MMFF94 forcefield, which is designed to simulate organic structures [81]. Since the structures of the PINK1 and GPR55 proteins were not crystallized for the *Homo sapiens*, they were obtained from the AlphaFold database. AlphaFold is an artificial intelligence program developed by DeepMind that predicts protein structures based on amino acid sequences. It demonstrates remarkable accuracy in predicting the three-dimensional conformations of proteins. By utilizing AlphaFold, we were able to access reliable structural models for PINK1 and GPR55.

Molecular docking simulations were carried out using AutoDock v.4.2. [82], and Vina [83]. The docking procedure was used to explore the ligand–receptor interactions at the PINK-1 and GPR55 binding site. For PINK1, a three-dimensional grid box was used, centered at X = 7.922, Y = 3.074, and Z = −5.849, with dimensions of 38.25 × 38.25 × 38.25 Å and a spacing of 0.375 Å. For GPR55, a three-dimensional grid box was used, centered at X = 7.231, Y = 2.315, and Z = −5.564, with dimensions of 84 × 84 × 84 Å and a spacing of 0.375 Å. The compounds were classified via the exponential consensus ranking (ECR) procedure [84] using the Autodock, Vina, and Vinardo [85] scoring functions. For the ligand conformational search in Autodock, the Lamarckian genetic algorithm (LGA) was used, and the parameters were set as follows: energy evaluations: 25,000,000; population size: 150; cross-over rate: 0.8; docking trials: 10; and mutation rate: 0.02. Random starting coordinates and orientations were used, with a clustering tolerance of 2.0 Å. A local search was conducted with a maximum of 300 iterations and a local search rate of 0.06. For the ligand conformational search in Vina, the exhaustiveness of the search was set to 64, allowing for 50 binding modes to be generated; the scoring functions used were vinardo and vina. Cisplatin was not evaluated using the Vina and Vinardo scoring functions, as these scoring methods are not optimized for platinum-containing atoms.

#### 4.5.2. ADMET/Pharmacokinetic Predictions and Transcriptomic Data

In the present study, various online web tools/software, including SwissADME (http://www.swissadme.ch/index.php, accessed on 14 September 2024), admetSAR 2.0 (http://lmmd.ecust.edu.cn/admetsar2, accessed on 10 October 2025), and pkCSM (http://biosig.unimelb.edu.au/pkcsm/prediction, accessed on 10 October 2025), were employed to predict ADMET/pharmacokinetics properties and to profile physicochemical parameters [86,87,88,89]. Additionally, data mining across multiple transcriptomic and expression profiling databases, such as UALCAN, GLIOVIS, GEPIA, and the Human Protein Atlas (https://www.proteinatlas.org/), was utilized to enhance the understanding of molecular characteristics and expression profiles [90,91,92,93,94,95].

### 4.6. Cell Viability Assays

#### MTT Assay

The MTT assay was performed to assess the viability of cells treated with cannabinoid compounds, black pepper extracts, and reference drugs (TMS, CisPt, and OSI). Cells were seeded at a density of 5 × 10^3^ cells per well in a 96-well plate and incubated overnight at 37 °C in a 5% CO_2_ atmosphere to allow for cell adherence. For the assay, cells in the presence and absence of treatments were incubated in 96-well plates for 12 h at 37 °C. Subsequently, 0.83 mg/mL of MTT (bromide of 3(4,5 dimethylthiazol) 2,5 diphenyl tetrazolium) was added, and the cells were incubated for 4 h at 37 °C to allow formazan crystal formation. After incubation, the unreacted MTT was removed, and dimethyl sulfoxide (DMSO) was added to dissolve the crystals. Each well was read at an absorbance of 540 nm using a spectrophotometer. Each evaluation of the compounds was performed at least in triplicate, and the data were analyzed using Magellan™ tecan v7.2 software to generate absorbance values and transform them into cell viability percentages. Cell viability was calculated as a percentage relative to untreated control wells, with the results used to establish IC_50_ values for each compound through dose–response curve analysis.

### 4.7. Mitochondrial Potential and Immunofluorescence Assay

To assess the mitochondrial membrane potential (ΔΨm), cells are first incubated with MitoTracker dyes, which are lipophilic, cationic fluorescent probes that accumulate in the mitochondria in a membrane-potential-dependent manner. We started by growing the cells in appropriate culture conditions until they reach the desired confluence. Afterward, the culture medium was removed, and the cells with PBS were washed to remove any residual medium. A working solution of MitoTracker dye (usually MitoTracker Green or MitoTracker Orange, depending on the desired fluorescence) was prepared at the recommended concentration, typically 50–200 nM, in fresh, pre-warmed culture medium without serum. The dye was added to the cells and incubated at 37 °C for 30 min to 1 h. During incubation, the dye enters the cells and accumulates within the mitochondria, where it becomes more fluorescent as the membrane potential is maintained. After incubation, the dye solution was removed, the cells were washed twice with PBS to eliminate any unbound dye, and we proceeded with fluorescence microscopy or flow cytometry analysis. The fluorescence intensity will correlate with the mitochondrial membrane potential, with a higher intensity indicating a more polarized mitochondrial membrane. Alternatively, a reduction in fluorescence intensity may indicate a loss of mitochondrial membrane potential, a common marker of mitochondrial dysfunction [96,97].

For the IF, cells were seeded in a 6-well plate, where sterilized glass coverslips were previously placed. In total, 100 μL of the fixation solution (4% (*w*/*v*) glutaraldehyde, 4% (*w*/*v*) sucrose in 1x PBS, with 1 mM calcium and magnesium) was added to each well and incubated for 15 min. Two washes were performed with 1x PBS for 5 min each at room temperature, and the solution was removed. Next, the cells were permeabilized using a solution of Tween 20 (100 μL per well) and incubated at room temperature for 20–30 min. The solution was then removed, and two washes with 1x PBS were performed for 5 min each. Following this, 100 μL of the blocking solution (TPBS: 0.1% (*v*/*v*) Tween 20 in 1x PBS) was added and incubated at room temperature for 1–2 h without agitation. After incubation, the blocking solution was removed, and the wells were washed with 1x PBS. For primary staining, 50–100 μL of primary antibody solution (mouse) was added and incubated overnight at 4 °C in a humidified chamber. After incubation, the solution was removed, and three washes with TPBS 1x were performed for 5 min each. For secondary staining, 50–100 μL of the secondary antibody solution (anti-mouse 488) was added and incubated for 1–2 h at room temperature without shaking in a humidified chamber protected from light. After incubation, two washes with TPBS 1x were performed for 5 min each. Next, 50–100 μL of phalloidin solution was added, and the cells were incubated at room temperature for 20 min. The solution was removed, and two washes with 1x PBS were performed. Finally, 100 μL of the DAPI solution (1–5 mM) was added, and the cells were incubated for 10 min. Afterward, two washes with 1x PBS were performed [98]. The antibodies used included PINK1 (EPR20730), GPR55 (ab174700), and CNR1 (EPR23934–20) from Abcam.

### 4.8. Statistical Analysis

Statistical analysis and data visualization were conducted using GraphPad Prism (version 9). Data are expressed as the mean ± standard deviation. To evaluate the significance among different groups, one-way analysis of variance (ANOVA) was performed, followed by Dunnett’s post-hoc test. A *p*-value of less than 0.05 was considered statistically significant, with significance levels indicated as follows: * *p* ≤ 0.05, ** *p* ≤ 0.01, and *** *p* ≤ 0.001.

### 4.9. Ethics Statement

This project is covered under the EVALUATION ACT: No. 023-213 of 2021, issued by the Ethics Committee of the Vice-Deanship of Research and Extension of the Faculty of Medicine at the Universidad Nacional de Colombia, Sede Bogotá.

## 5. Conclusions

The molecular docking study highlights CBC, P, and THC as top candidates for modulating PINK1 and GPR55. While docking studies suggest strong interactions between *Piper nigrum* derivatives, cannabinoids, and targets such as PINK1 and GPR55, in vitro experiments confirmed the cytotoxic potential of these compounds in glioblastoma cell lines, with cannabinoids like CBG and CBD showing significant dose-dependent reductions in cell viability, comparable to established chemotherapeutic agents. Interestingly, the addition of *Piper nigrum* derivatives to cannabinoid treatments revealed a complex interaction, particularly in the CCF-STTG1 cell line, where *Piper nigrum* seemed to counteract the cytotoxic effects of cannabinoids, suggesting potential modulatory effects on PINK1 activity. This may provide a novel way for personalized treatment strategies in GBM depending on the specific tumor context.

Furthermore, transcriptomic and survival data from public databases underscored the potential of PINK1 as a critical biomarker in glioblastoma. PINK1’s downregulation in GBM tissues, along with its complex role in mitochondrial homeostasis and energy metabolism, suggests its potential as both a therapeutic target and prognostic indicator. While GPR55 did not show a significant correlation with GBM prognosis in this study, our group is investigating its potential expression profile in patient samples and the possible molecular effects of cannabinoid treatments to clarify these inconsistencies and resolve discrepancies between the literature and various databases. Additionally, its role in other cancers warrants continued investigation.

Overall, the integration of molecular docking, transcriptomic analysis, and cytotoxicity testing presents a promising foundation for the development of cannabinoid-based therapies in glioblastoma. Given the challenges of current treatments and the limited survival rates of GBM patients, plant-derived compounds offer a potential alternative or adjunctive therapy. Future clinical trials are essential to validate these findings and explore the therapeutic utility of cannabinoids and *Piper nigrum* derivatives in glioma treatment, potentially improving patient outcomes and survival. This underscores the need for integrated approaches combining computational, biochemical, and cellular models to achieve a more comprehensive understanding. Future studies addressing these discrepancies could lead to optimized therapeutic strategies tailored to the molecular and cellular environment of specific tumor types.

Interestingly, this study is the first to show the modulation of PINK1 by cannabinoids as a potential mechanism underlying their cytotoxic effects in cancer. At least in part, this is influenced by the complex pharmacology of cannabinoids. This discovery opens the door to new mechanisms of action in oncology, suggesting that PINK1 modulation could enhance chemotherapy [99], radiotherapy [100], and other therapeutic approaches [67,68]. However, it may also have detrimental effects, particularly in immunotherapy, due to its potential role in PD-L1 regulation [76]. Notably, cannabinoids have been associated with poorer immunotherapy outcomes [101], which could, at least partially, be explained by their influence on PINK1. Further studies are needed to evaluate this.

## Figures and Tables

**Figure 1 ijms-26-05688-f001:**
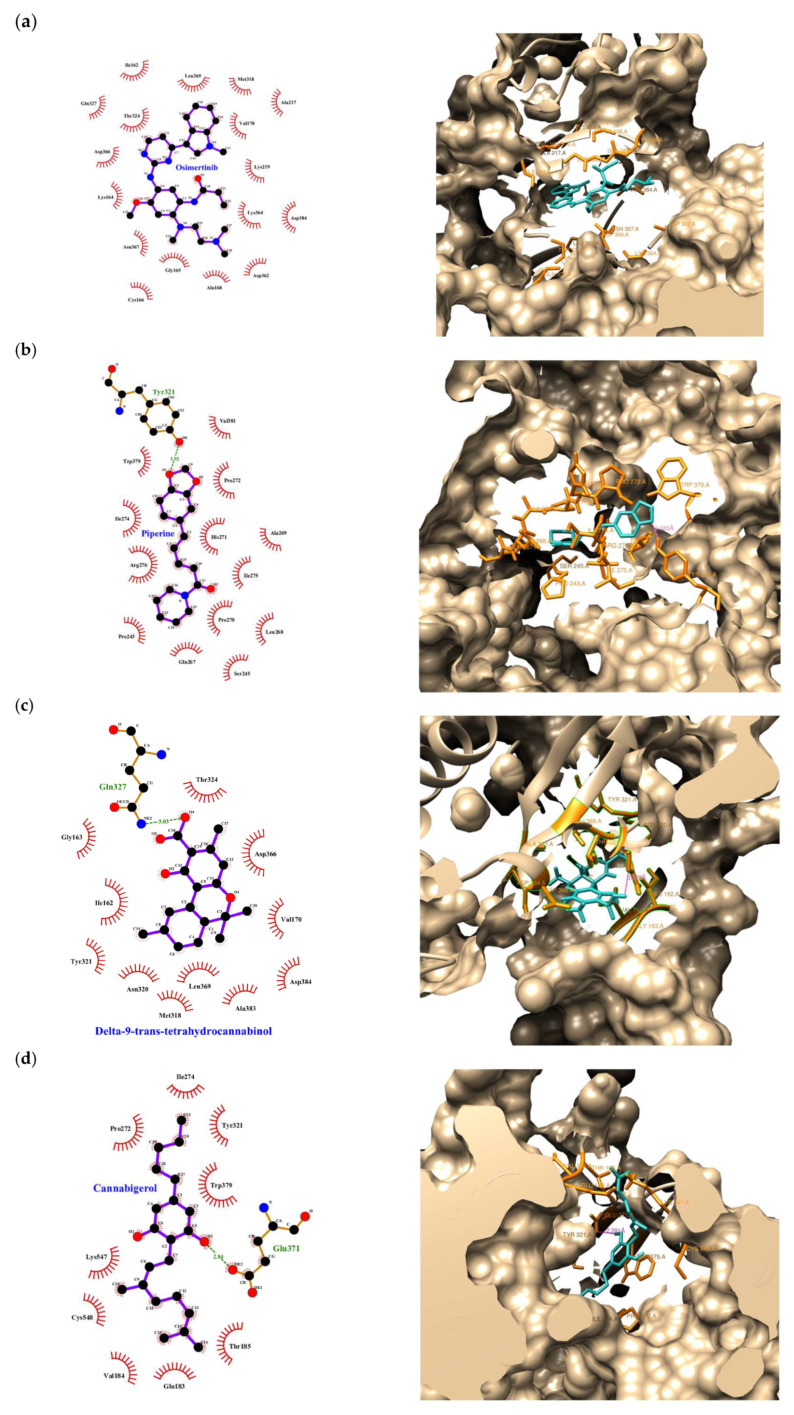
Binding Interactions of ligands with PINK1. (**a**) OSI, (**b**) P, (**c**) THC, (**d**) CBG, (**e**) CBDA, (**f**) CBC, (**g**) CBD, (**h**) BCP. These ligands engage critical residues involved in PINK1’s catalytic activity, suggesting potential for targeted therapies. OSI binds strongly via hydrophobic and electrostatic interactions with key residues like Ile162, Leu369, Met318, Asp366, and Lys164. P forms a hydrogen bond with Tyr321. THC binds through hydrogen bonds with Gln327 and hydrophobic interactions with Val170 and Asp366. CBG and CBDA interact with Glu371, Trp379, and Val184.

**Figure 2 ijms-26-05688-f002:**
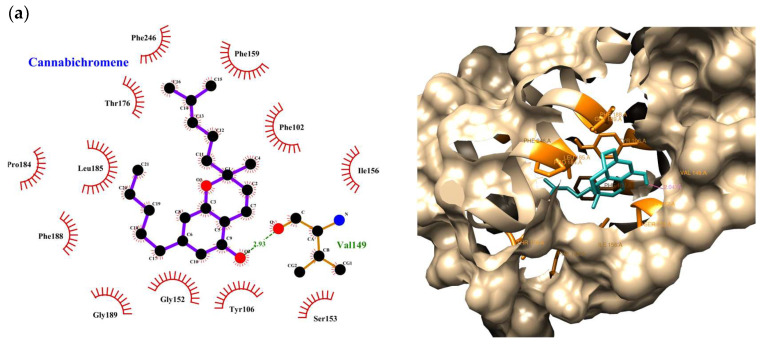
Binding interactions of ligands with GPR55. (**a**) CBC, (**b**) OSI, (**c**) P, (**d**) CBG, (**e**) CBDA, (**f**) THC, (**g**) CBD, (**h**) BCP. Hydrophobic interactions are observed with residues such as Phe102, Phe159, and Leu148 for ligands like CBC, CBG, CBD, THC, OSI, P, and BCP. Hydrogen bonds are formed between specific residues and ligands like CBC, OSI, THC, and CBDA. Ligands such as P, CBG, CBD, and BCP primarily interact through hydrophobic contacts.

**Figure 3 ijms-26-05688-f003:**
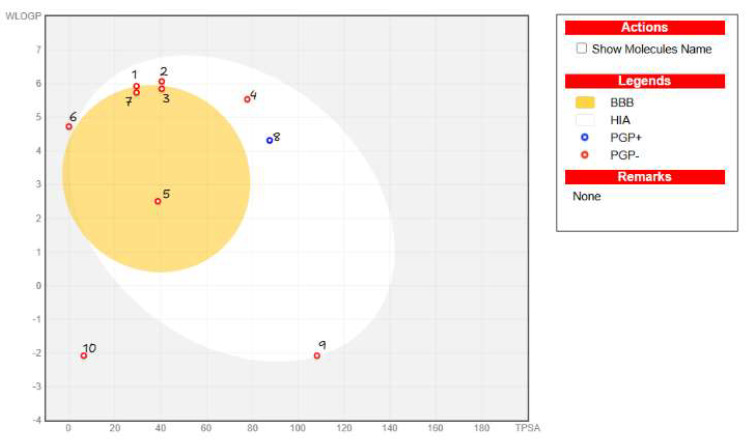
Boiled egg diagram for all compounds analyzed. 1. CBC. 2. CBG. 3. CBD. 4. CBDA. 5. Piperine. 6. BCP. 7. THC. 8. OSI. 9. TMZ. 10. CisPt. BBB: Blood–brain barrier. HIA: Human intestinal absorption. PgP: P-glycoprotein.

**Figure 4 ijms-26-05688-f004:**
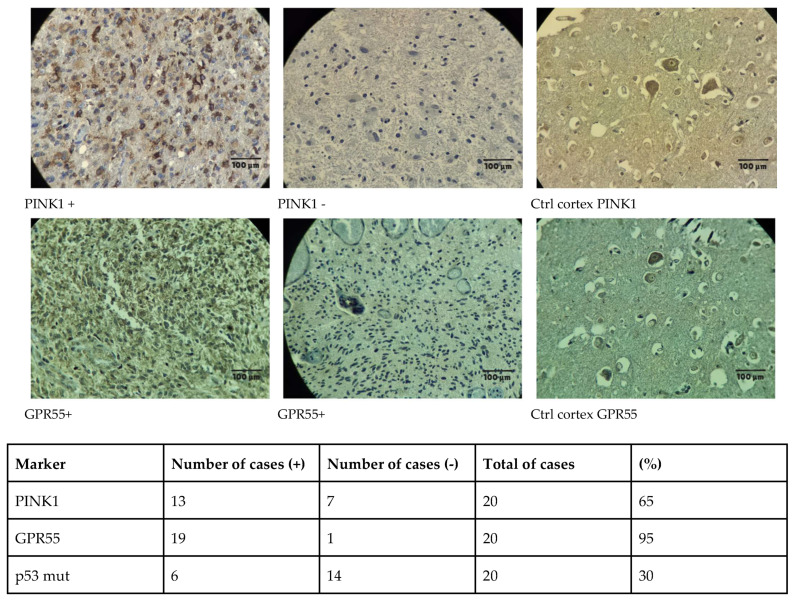
Immunohistochemical analysis of PINK1 and GPR55 expression in glioblastoma patient samples.

**Figure 5 ijms-26-05688-f005:**
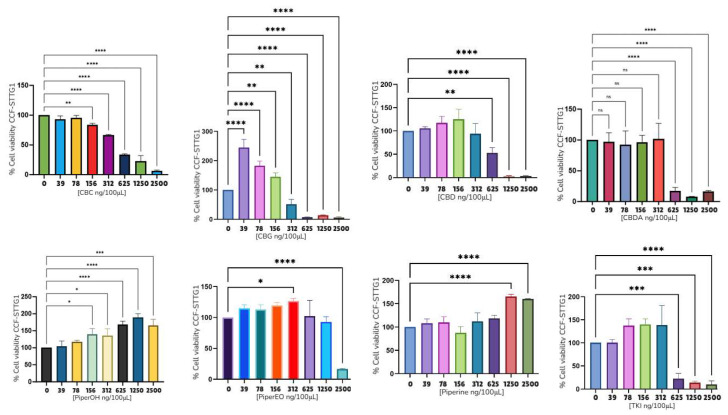
MTT viability assays of CCF-STTG1, 24 h. The asterisks in the figure represent the following levels of statistical significance: *p* < 0.05 (*), *p* < 0.01 (**), *p* < 0.001 (***), *p* < 0.0001 (****).

**Figure 6 ijms-26-05688-f006:**
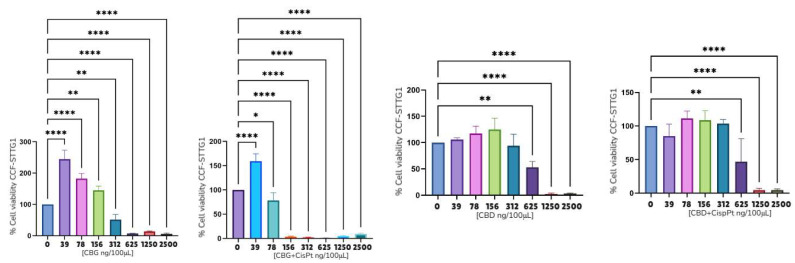
MTT viability assays of CBG and CBD with CisPt in CCF-STTG1. The asterisks in the figure represent the following levels of statistical significance: *p* < 0.05 (*), *p* < 0.01 (**), *p* < 0.0001 (****).

**Figure 7 ijms-26-05688-f007:**
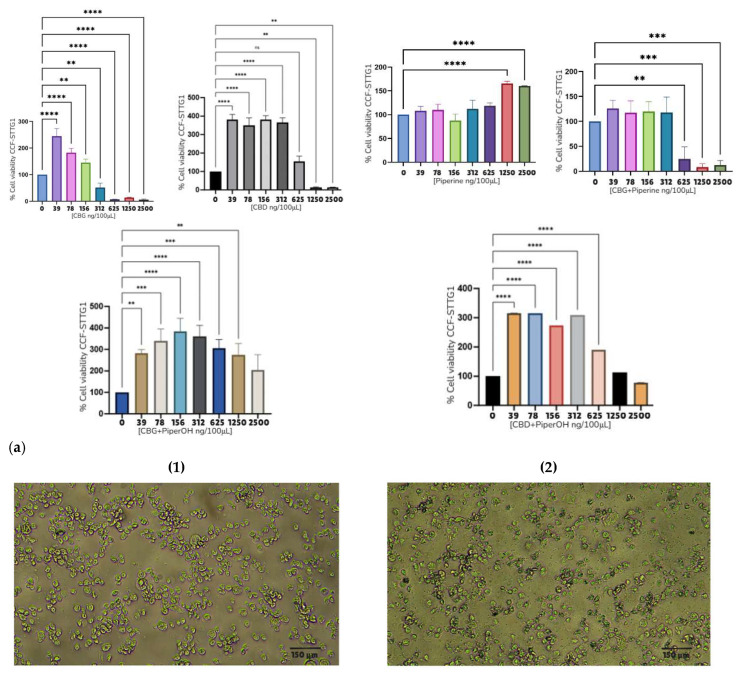
Viability assays and morphological changes in the CCF-SSTG1 cell line treated with selected compounds. (**a**) MTT viability assays of CBD, CBG, PiperOH, and P in CCF-STTG1. (**b**) Morphological changes in the CCF-SSTG1 cell line treated with CBD, CBG, and CBG + POH, observed under an optical microscope at 10× magnification with a scale of 150 µm. 1. CBD 25 ng/μL; 2. CBD and POH 25 ng/μL each; 3. CBG 25 ng/μL; 4. CBG + POH 25 ng/μL each; 5. control. Note: 2500 ng/100 μL = 25 ng/μL. The asterisks in the figure represent the following levels of statistical significance: *p* < 0.01 (**), *p* < 0.001 (***), *p* < 0.0001 (****).

**Figure 8 ijms-26-05688-f008:**
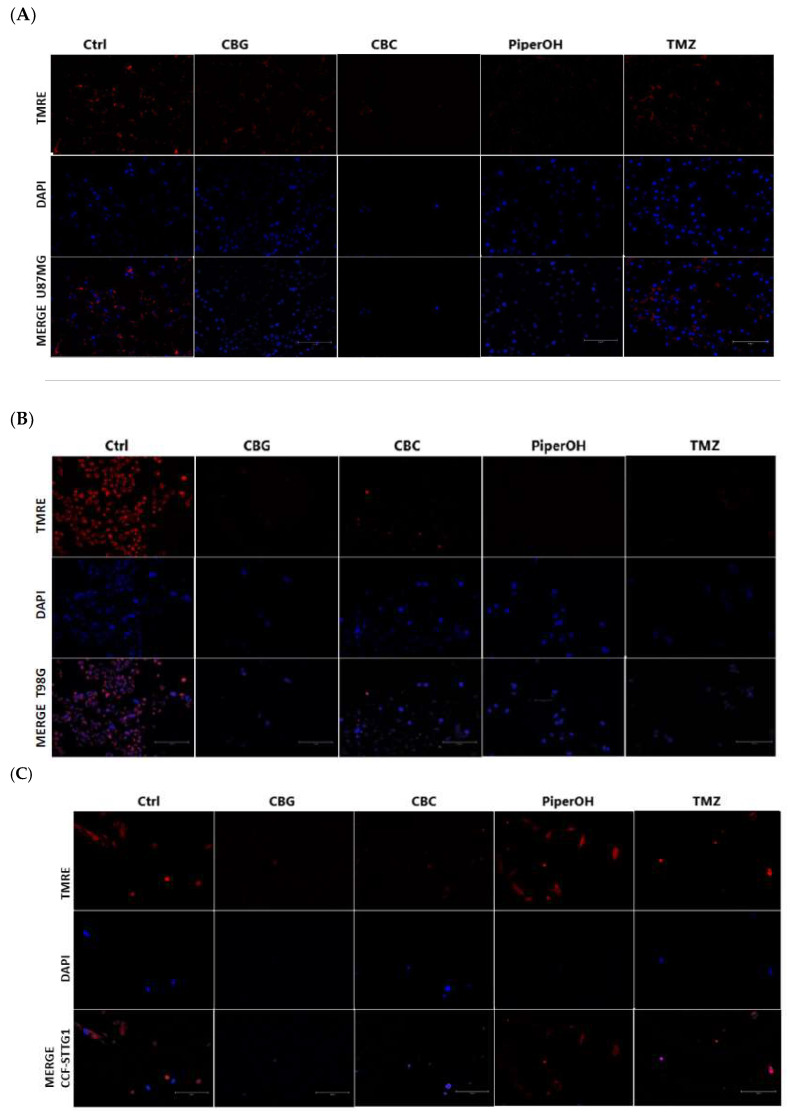
Assessment of mitochondrial membrane potential in astrocytoma cells U87 mg, T98G, and CCFSTTG1. (**A**–**C**) Cells were treated with control (Ctrl), CBG, CBC (4 ng/µL), PiperOH (12.5 ng/µL), and TMZ (50 ng/µL) for 24 h and analyzed via confocal microscopy (magnification × 20 scale bar 150 µm) using TMRE dye, with nuclear staining using DAPI.

**Figure 9 ijms-26-05688-f009:**
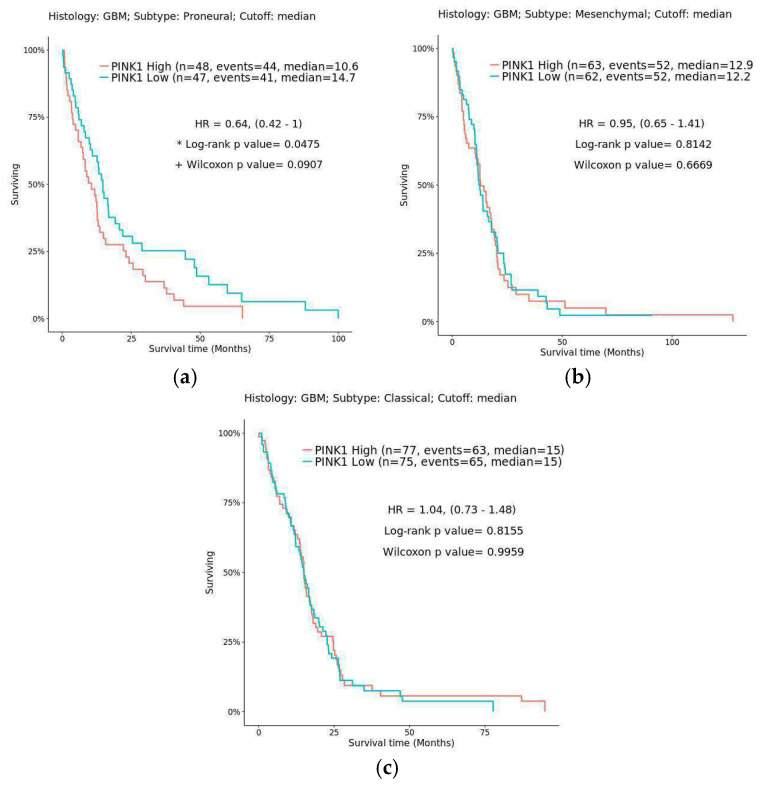
Kaplan–Meier estimator survival analysis of the glioblastoma subtype correlated with PINK1 expression: (**a**) proneural, (**b**) mesenchymal, and (**c**) classical. The blue line indicates low PINK1 expression, and the lilac line indicates high PINK1 expression. Statistical significance is indicated by * for the log-rank test *p*-value and + for the Wilcoxon test *p*-value. A *p*-value < 0.05 is considered statistically significant for both tests. The log-rank test assesses overall survival differences between groups, while the Wilcoxon test emphasizes early survival differences.

**Figure 10 ijms-26-05688-f010:**
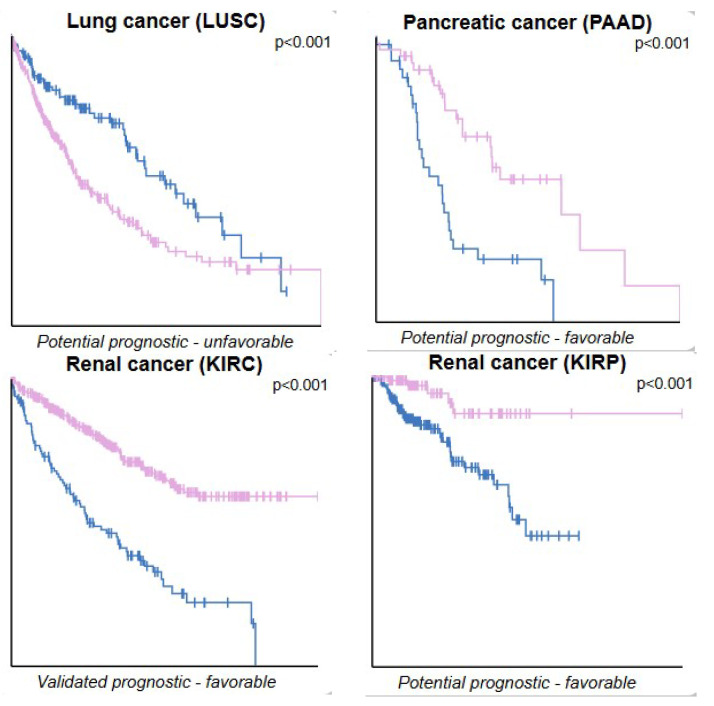
PINK1 as a prognostic marker in other types of cancer. Kaplan–Meier plots illustrate the survival outcomes for various cancers where high PINK1 expression levels show a statistically significant correlation (*p* < 0.001) with patient survival. The prognosis—whether favorable or unfavorable—is noted in brackets alongside each plot. Survival analysis: Kaplan–Meier plots summarize results from analysis of correlations between mRNA expression levels and patient survival. Patients were divided, based on expression levels, into one of the two following groups: “low” (under cut-off) or “high” (over cut-off). The x-axis shows the time for survival (years), and the y-axis shows the probability of survival, where 1.0 corresponds to 100 percent. The blue line indicates low PINK1 expression, and the lilac line indicates high PINK1 expression.

**Figure 11 ijms-26-05688-f011:**
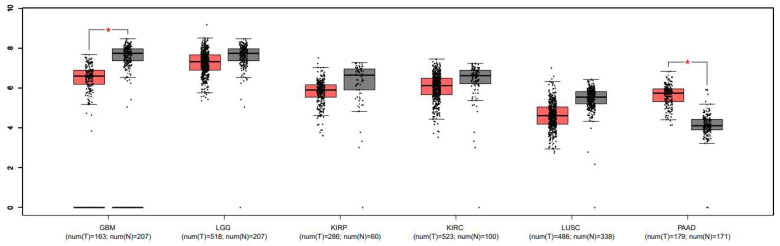
Boxplot comparing the expression levels of PINK1 between tumor (T) and normal (N) tissues across different cancer types. Red indicates tumor samples, and gray represents normal tissue. Significant differences in expression are marked with an asterisk (*). Cancer types include glioblastoma multiforme (GBM), low-grade glioma (LGG), kidney renal papillary cell carcinoma (KIRP), kidney renal clear cell carcinoma (KIRC), lung squamous cell carcinoma (LUSC), and pancreatic adenocarcinoma (PAAD). Expression− log2(TPM + 1).

**Figure 12 ijms-26-05688-f012:**
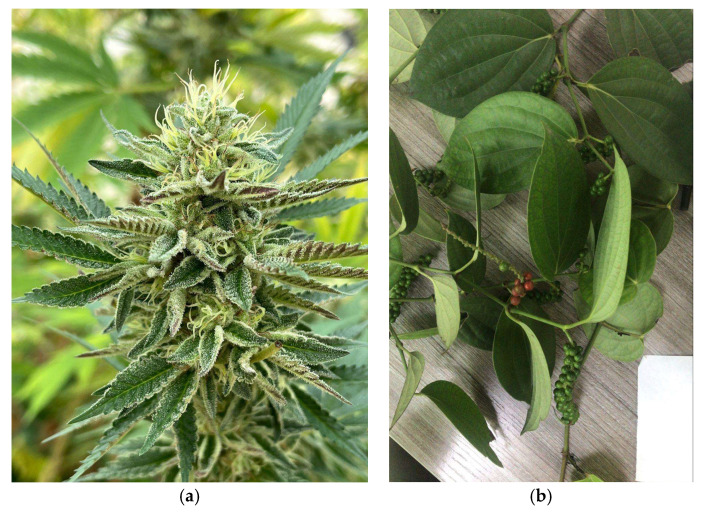
Specimens of *Cannabis sativa* (**a**) and *Piper nigrum* (**b**).

**Table 2 ijms-26-05688-t002:** Docking affinities and ECR scores for compounds binding to PINK1.

Compound	Rank Vina	Vina Affinity kcal/mol	Rank Vinardo	Vinardo Affinity kcal/mol	Rank Autodock	Autodock Affinity kcal/mol	ECR
Osimertinib (OSI)	1	−9.30	3	−9.00	2	−7.49	0.53
Piperine (P)	3	−8.00	6	−7.80	1	−9.75	0.41
Tetrahydrocannabinol (THC)	2	−8.70	7	−7.70	3	−7.35	0.33
Cannabigerol (CBG)	8	−7.00	1	−9.20	7	−6.86	0.29
Cannabidiolic acid (CBDA)	6	−7.40	2	−9.10	9	−6.37	0.23
Cannabichromene (CBC)	5	−7.50	4	−8.40	5	−7.04	0.21
Cannabidiol (CBD)	4	−7.60	5	−8.10	8	−6.62	0.17
β-Caryophyllene (BCP)	7	−7.20	8	−5.50	4	−7.14	0.14

**Table 3 ijms-26-05688-t003:** Docking affinities and ECR scores for compounds binding to GPR55.

Compound	Rank Vina	Vina Affinity kcal/mol	Rank Vinardo	Vinardo Affinity kcal/mol	Rank Autodock	Autodock Affinity kcal/mol	ECR
Cannabichromene (CBC)	1	−9.20	6	−7.30	2	−9.40	0.46
Osimertinib (OSI)	6	−7.90	3	−8.60	1	−9.46	0.41
Piperine (P)	2	−8.50	5	−7.80	3	−8.36	0.36
Cannabigerol (CBG)	3	−8.30	2	−8.80	5	−7.81	0.36
Cannabidiolic acid (CBDA)	7	−7.70	1	−9.50	7	−7.73	0.30
Tetrahydrocannabinol (THC)	4	−8.20	7	−7.10	4	−7.96	0.21
Cannabidiol (CBD)	5	−8.10	4	−8.50	6	−7.77	0.20
β-Caryophyllene (BCP)	8	−7.50	9	−5.80	8	−7.26	0.06

**Table 4 ijms-26-05688-t004:** In silico toxicity profiles of compounds analyzed.

	Compounds Analyzed								
Toxicity properties	CBC	CBG	CBD	CBDA	P	BCP	THC	OSI	TMZ
AMES toxicity	No	No	No	No	No	No	No	No	Yes
hERG I inhibitor	No	No	No	No	No	No	No	No	No
Carcinogenicity	No	No	No	No	No	No	No	No	Yes
Hepatotoxicity	No	No	No	No	Yes	No	No	Yes	Yes
Skin sensitization	No	Yes	No	No	No	Yes	No	No	No

AMES toxicity: Test for mutagenic potential using bacteria. hERG I inhibitor: Inhibits human Ether-à-go-go-Related Gene (hERG) potassium channels, potentially causing arrhythmias. Carcinogenicity: Potential to cause cancer. Hepatotoxicity: Toxicity to the liver. Skin sensitization: Potential to cause allergic skin reactions.

**Table 5 ijms-26-05688-t005:** IC_50_ values of selected compounds in various cancer cell lines expressed in ng/μL and μM.

	Compounds Analyzed						
Cell Line	CBC	CBG	CBD	CBDA	PiperOH	PiperEO	P
U87MG	470 ng/100 μL	470 ng/100 μL	470 ng/100 μL	938 ng/100 μL	1250 ng/100 μL	—	—
	4.7 ng/μL	4.7 ng/μL	4.7 ng/μL	9.3 ng/μL	12.5 ng/μL	—	—
	—	14.8 µM	14.9 µM	—	—	—	—
T98G	312 ng/100 μL	156 ng/100 μL	156 ng/100 μL	—	625 ng/100 μL	—	—
	3.12 ng/μL	1.56 ng/μL	1.56 ng/μL	—	6.25 ng/μL	—	—
	—	4.93 μM	4.96 μM	—	—	—	—
CCF-STGG1	312 ng/100 μL	312 ng/100 μL	625 ng/100 μL	470 ng/100 μL	NC*	2500 ng/100 μL	NC*
	3.12 ng/μL	3.12 ng/μL	6.25 ng/μL	4.70 ng/μL	NC*	25 ng/μL	NC*
	—	9.91 μM	19.87 μM	—	NC*	—-	NC*
Mean ± S.E.M. Astrocytoma cell lines	—	9.88 ± 2.8 μM	13.24 ± 4.3 μM	—	—	—-	—
SH-SY5Y	625 ng/100 μL	312 ng/100 μL	940 ng/100 μL	312 ng/100 μL	1000 ng/100 μL	940 ng/100 μL	—
	6.25 ng/μL	3.12 ng/μL	9.4 ng/μL	3.12 ng/μL	10 ng/μL	9.4 ng/μL	—
	—	9.91 μM	29.9 μM	—	—	—	—
MO3.13	156 ng/100 μL	312 ng/100 μL	625 ng/100 μL	312 ng/100 μL	2500 ng/100 μL	—	—
	3.12 ng/μL	3.12 ng/μL	6.25 ng/μL	3.12 ng/μL	25 ng/μL	—	—
	—	9.91 μM	19.87 μM	—	—	—	—

NC*: Non-cytotoxic.

**Table 6 ijms-26-05688-t006:** Data used for the Kaplan–Meier plots of Figure 10 for various cancers with different expression levels of PINK1.

Type of Cancer	n Low-Expression mRNA (Blue)	n High-Expression mRNA (Lilac)	*p* Score	Median Follow-Up Time (Years)	5-Year Survival Low	5-Year Survival High
Lung squamous cell carcinoma	137	352	0.00027	1.81	70%	39%
Pancreatic Adenocarcinoma	36	44	0.00014	1.15	0%	12%
Kidney Renal Clear Cell Carcinoma	114	407	3.7 × 10^−14^	3.29	38%	71%
Kidney Renal Papillary Cell Carcinoma	182	100	0.001	2.11	69%	87%

## Data Availability

Data is contained within the article and Appendix A.

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
