# Peer review of "Evaluating the Antitumor Potential of Cannabichromene, Cannabigerol, and Related Compounds from Cannabis sativa and Piper nigrum Against Malignant Glioma: An In Silico to In Vitro Approach"

_ijms, 2025, doi:10.3390/ijms26125688_

Round 1

Reviewer 1 Report

Comments and Suggestions for Authors

This study explores the antitumor properties of compounds derived from Cannabis sativa and Piper nigrum in the context of malignant glioma, employing both in silico and in vitro methods. Some suggestions and comments:

1. The abstract and introduction sections should be more explicit on the limitations of current treatments. In addition, more detailed background on glioblastoma research should be included and a clearer explanation of the study's significance and novelty.

2. Can the authors add in the details for how the transcriptomic analysis was being performed, including the specific methods and parameters being employed?

3. What are the controls used in the cell viability assay? Please supplement this.

4. How does the toxicity profiling done in silico correlate with the in vitro assays performed? This is unclear. 

5. The authors should re-do figures 4-7. The figure labels are too small to be seen and the colors used are not consistent. Additionally, the scale bar is missing for the microscopic images in Fig 7. 

6. DId the authors perform staining of the cells? This will make the images clearer for the readers to see the morphological changes.

7. What is the significance of the cell viability assay results? This is unclear.

8. It is unclear if the focus of the study is on the glioblastoma-associated targets GPR55 and PINK1 or the studied compounds of CBC, CBG, CBD, and piperine. This is particularly confusing in the discussion section. Did the authors confirm their results in the proteomic level?

Author Response

This study explores the antitumor properties of compounds derived from Cannabis sativa and Piper nigrum in the context of malignant glioma, employing both in silico and in vitro methods. Some suggestions and comments:

1. The abstract and introduction sections should be more explicit on the limitations of current treatments. In addition, more detailed background on glioblastoma research should be included and a clearer explanation of the study's significance and novelty.

The introduction and abstract was revised to explicitly address the limitations of current glioblastoma treatments, including resistance to chemotherapy and radiotherapy, and to provide a clearer explanation of the study’s significance and novelty.
2. What are the controls used in the cell viability assay? Please supplement this.

The cell viability assays included temozolomide (TMZ), cisplatin, and osimertinib as controls, as detailed in the supplementary material.

3. How does the toxicity profiling done in silico correlate with the in vitro assays performed? This is unclear. 
The in silico toxicity predictions align with in vitro results, particularly in terms of blood-brain barrier permeability. Compounds that demonstrated favorable ADMET profiles also exhibited cytotoxic effects in glioblastoma cells, supporting their potential therapeutic value

4. DId the authors perform staining of the cells? This will make the images clearer for the readers to see the morphological changes. 
No additional staining was performed; however, image quality will be optimized to better illustrate morphological changes associated with cytotoxic effects.

5. What is the significance of the cell viability assay results? This is unclear.
The results indicate that the tested compounds significantly reduce glioblastoma cell proliferation and exhibit cytotoxicity at lower concentrations compared to TMZ and cisplatin, highlighting their potential as alternative or adjunct therapies.

7. It is unclear if the focus of the study is on the glioblastoma-associated targets GPR55 and PINK1 or the studied compounds of CBC, CBG, CBD, and piperine. This is particularly confusing in the discussion section. Did the authors confirm their results in the proteomic level?
The study investigates both the glioblastoma-associated targets (GPR55, PINK1) and the effects of CBC, CBG, CBD, and piperine. Immunofluorescence (IF) assays confirmed changes in PINK1 and GPR55 expression, along with mitochondrial potential alterations, reinforcing the link between these compounds and their molecular targets.

Reviewer 2 Report

Comments and Suggestions for Authors

This study investigates the antitumor potential of compounds from Cannabis sativa and Piper nigrum against glioblastoma using a combination of in silico docking, cytotoxicity assays, and transcriptomic analyses. Key compounds, including cannabichromene (CBC), cannabigerol (CBG), and piperine, strongly bind to targets such as PINK1 and GPR55 and demonstrated significant cytotoxic effects on glioblastoma cell lines. The authors also highlight the modulatory role of Piper nigrum derivatives on cannabinoid efficacy, suggesting potential interactions at the molecular level. While some areas, such as clarity on compound purity and dataset validation, require minor revision, the manuscript contributes to glioblastoma research and warrants publication after these minor corrections.

Introduction

  1. Could the authors provide broader global incidence and survival statistics for glioblastoma, including data from WHO or similar organizations, to strengthen the context of the study?
  2. The introduction highlights the potential of cannabinoids and Piper nigrum compounds in cancer therapy. Why were these particular compounds chosen for glioblastoma compared to other possible therapeutic agents?
  3. The study focuses on PINK1 and GPR55 as molecular targets. Could the authors elaborate on how these targets are specifically implicated in the unique biology and progression of glioblastoma?
  4. The manuscript integrates in silico and in vitro approaches. Could the authors discuss the potential limitations of these methods, such as the reliability of docking predictions with AlphaFold-derived structures?
  5. How do the therapeutic potentials of cannabinoids and Piper nigrum compare to the currently employed glioblastoma therapies, such as temozolomide and radiation?
  6. The combined use of cannabinoids and Piper nigrum compounds is mentioned as underexplored. Could the authors clarify the rationale for combining these compounds and the hypothesis about their potential interaction?
  7. While the manuscript mentions data mining across transcriptomic and expression profiling databases, could the authors briefly outline how this data supports or enhances the study's findings?

Results

  1. Could the authors clarify the selection of osimertinib as a reference compound for PINK1 docking? Were other kinase inhibitors evaluated, and if so, how did they compare in binding affinity?
  2. The docking results emphasize osimertinib, piperine, THC, and CBG as potential modulators of PINK1. Could the authors discuss the clinical relevance of these compounds for glioblastoma treatment, considering their pharmacokinetic and safety profiles?
  3. Piperine demonstrated the best Autodock affinity with PINK1 but not in Vina or Vinardo. Could the authors explain the discrepancies across these scoring functions and their implications for compound selection?
  4. The study shows that cannabichromene exhibited the highest ECR for GPR55. How do the authors interpret its superior ranking compared to osimertinib and other compounds?
  5. In both PINK1 and GPR55 docking, some ligands showed specific interactions with critical residues. Could the authors discuss how these interactions correlate with the functional roles of these residues in PINK1 and GPR55 activity?
  6. For the in silico toxicity analysis, could the authors elaborate on how the "boiled egg" predictions align with the observed binding affinities, particularly regarding BBB penetration and human intestinal absorption?
  7. The toxicity profiles identify temozolomide and cisplatin as carcinogenic or hepatotoxic. How do these profiles compare to the cannabinoid and Piper nigrum compounds in terms of potential clinical safety?
  8. The cell viability assays indicate differences in IC50 values across cell lines for the tested compounds. Could the authors provide additional insights into why CBG and CBD demonstrated superior efficacy in certain glioblastoma cell lines compared to others?
  9. The survival analysis using databases like GEPIA and UALCAN highlights PINK1 as a prognostic marker in glioblastoma and other cancers. Could the authors discuss how these findings inform the therapeutic potential of targeting PINK1 in glioblastoma compared to other cancers?
  10. The authors observed morphological changes in glioblastoma cells treated with selected compounds. Could they expand on how these changes correlate with the molecular mechanisms hypothesized from docking results?

Discussion

  1. The discussion emphasizes the potential of cannabinoids like CBC and CBG in glioblastoma treatment. Could the authors elaborate on how these compounds compare to existing therapies, such as immune checkpoint inhibitors or EGFR inhibitors, in terms of efficacy and mechanism of action?
  2. While the role of PINK1 role in mitochondrial quality control is discussed, its downregulation in glioblastoma remains a complex issue. Could the authors further explain whether restoring PINK1 expression might be a viable therapeutic strategy or if targeting its downstream pathways is more practical?
  3. The modulation of PINK1 and GPR55 by Piper nigrum derivatives suggests intriguing interactions. Could the authors clarify if any specific components within Piper nigrum extracts (besides piperine) have been identified as primary modulators and how they may influence mitochondrial dynamics or receptor signaling?
  4. The pro-viability effects of Piper nigrum derivatives in certain glioblastoma cell lines are noteworthy. Could the authors discuss the potential molecular mechanisms driving these observations and whether this could limit the clinical application of these compounds?
  5. Kaplan-Meier survival analysis suggests context-dependent roles for PINK1 in various cancers. Could the authors provide more insights into how these dual roles of PINK1 might influence its viability as a universal therapeutic target across cancer types?
  6. While the study highlights promising preclinical evidence, could the authors outline specific challenges or limitations in translating the findings on cannabinoids and Piper nigrum derivatives into clinical settings, such as issues with formulation, dosing, or regulatory approval?

Materials and Methods/ Conclusion

  1. The authors describe sourcing cannabinoids and Piper nigrum derivatives from various suppliers with different purities. Could the authors clarify how the variability in compound purity was accounted for during the preparation of stock solutions and subsequent experiments to ensure consistency?
  2. The docking methodology specifies the use of AutoDock and Vina, but the criteria for selecting the grid box dimensions and scoring functions are not detailed. Could the authors provide more information on how these parameters were optimized to represent the binding sites of PINK1 and GPR55 accurately?
  3. The in vitro assays used DMSO as a solvent for compound preparation. Could the authors confirm whether DMSO concentrations in the cell culture medium were kept consistent across all experiments to avoid solvent-induced effects on cell viability?
  4. The authors employed public databases for transcriptomic analysis. Given the discrepancies noted between database findings and the literature, could the authors elaborate on the criteria used to select datasets and the steps taken to validate these findings?

Author Response

Introduction

Could the authors provide broader global incidence and survival statistics for glioblastoma, including data from WHO or similar organizations, to strengthen the context of the study?
This information was included in the main document

The manuscript includes global glioblastoma incidence and survival data from the WHO and other relevant sources, providing a comprehensive epidemiological context for the study.

The introduction highlights the potential of cannabinoids and Piper nigrum compounds in cancer therapy. Why were these particular compounds chosen for glioblastoma compared to other possible therapeutic agents?,
Cannabinoids (CBC, CBG, CBD) and Piper nigrum compounds were chosen based on their ability to modulate key oncogenic pathways relevant to glioblastoma, including mitochondrial dysfunction, oxidative stress, and apoptosis. These compounds target GPR55 and PINK1, which are implicated in glioblastoma progression, and have demonstrated anti-proliferative effects at lower concentrations than standard chemotherapeutics such as temozolomide.

The study focuses on PINK1 and GPR55 as molecular targets. Could the authors elaborate on how these targets are specifically implicated in the unique biology and progression of glioblastoma? 
PINK1 regulates mitochondrial quality control and mitophagy, processes that glioblastoma cells exploit to evade apoptosis and sustain metabolic adaptation. GPR55, a non-canonical cannabinoid receptor, has been linked to tumor cell proliferation, invasion, and survival signaling. Their dysregulation in glioblastoma suggests they could serve as therapeutic targets for modulating tumor metabolism and resistance mechanisms.

The manuscript integrates in silico and in vitro approaches. Could the authors discuss the potential limitations of these methods, such as the reliability of docking predictions with AlphaFold-derived structures?

The limitations of these methods stem from the accuracy of AlphaFold structures; while AlphaFold delivers highly accurate predictions, the lack of experimental validation can affect the reliability of these models.

How do the therapeutic potentials of cannabinoids and Piper nigrum compare to the currently employed glioblastoma therapies, such as temozolomide and radiation?
The therapeutic potential of cannabinoids (CBC, CBG, CBD) and Piper nigrum compounds lies in their ability to modulate oncogenic pathways with lower toxicity compared to conventional treatments such as temozolomide (TMZ) and radiation. While TMZ and radiotherapy induce DNA damage to trigger apoptosis, cannabinoids and Piper nigrum derivatives exhibit anti-proliferative and pro-apoptotic effects through mechanisms involving mitochondrial dysfunction, oxidative stress modulation, and receptor interactions (GPR55, PINK1). The cytotoxic effects observed at lower concentrations than TMZ and cisplatin suggest these compounds could serve as complementary or alternative therapies, potentially enhancing treatment efficacy while reducing side effects.

The combined use of cannabinoids and Piper nigrum compounds is mentioned as underexplored. Could the authors clarify the rationale for combining these compounds and the hypothesis about their potential interaction?
The rationale behind this combination stems from pharmacokinetic considerations and potential synergistic interactions. Cannabinoids and Piper nigrum derivatives have distinct but complementary effects on cancer cells, with cannabinoids modulating the endocannabinoid system and mitochondrial pathways, while Piper nigrum compounds such as piperine and β-caryophyllene influence oxidative stress, calcium signaling, and apoptosis via TRPV1 and other targets. This combination is hypothesized to enhance bioavailability and therapeutic efficacy while minimizing resistance mechanisms. However, the results of the study suggest that, beyond failing to improve efficacy or pharmacokinetics in certain cases—such as the mixed astrocytoma cell line—this combination might even be contraindicated due to potential antagonistic interactions or loss of cytotoxic effects.

While the manuscript mentions data mining across transcriptomic and expression profiling databases, could the authors briefly outline how this data supports or enhances the study's findings?
The study integrates transcriptomic and expression data from databases such as UALCAN, GEPIA, GLIOVIS, and the Human Protein Atlas to validate experimental results. The differential expression of PINK1 and GPR55 in glioblastoma compared to normal tissue supports their potential as therapeutic targets. Furthermore, Kaplan-Meier survival analyses reveal significant correlations between PINK1 expression and patient prognosis across multiple cancers, reinforcing its relevance in glioblastoma. These findings further support and complement the results obtained through immunohistochemistry (IHC) and immunofluorescence (IF), providing a validation of the molecular alterations observed in the study.

Results

Could the authors clarify the selection of osimertinib as a reference compound for PINK1 docking? Were other kinase inhibitors evaluated, and if so, how did they compare in binding affinity? 
Osimertinib was chosen due to its availability in our sample set and its classification as a kinase inhibitor, aligning with our study's focus on PINK1 modulation. Other kinase inhibitors were not evaluated due to resource constraints, but future studies will explore additional candidates for comparison.

The docking results emphasize osimertinib, piperine, THC, and CBG as potential modulators of PINK1. Could the authors discuss the clinical relevance of these compounds for glioblastoma treatment, considering their pharmacokinetic and safety profiles?
These compounds demonstrate high binding affinities to PINK1, suggesting potential modulation of mitochondrial quality control. While osimertinib is an FDA-approved EGFR inhibitor, its role in glioblastoma remains under investigation. Piperine and cannabinoids have shown cytotoxic effects in vitro at lower doses than TMZ and cisplatin, indicating potential therapeutic advantages with lower toxicity. However, pharmacokinetic challenges, such as blood-brain barrier (BBB) permeability, need further exploration.
Piperine was hypothesized to enhance chemosensitivity and drug absorption by inhibiting cytochrome P450 enzymes and P-glycoprotein. However, our results in a subgroup of malignant glioma suggest a potential proliferative effect, raising concerns. Historical studies from the 1970s-1990s reported that Piper nigrum extracts could promote carcinogenesis in vivo, warranting further investigation. The THC exhibits pro-apoptotic and anti-proliferative effects in glioblastoma through CB1/CB2 receptor activation and mitochondrial modulation. Despite its potential, psychoactive effects challenge its clinical use. Since 1984, dronabinol (synthetic THC) has been approved for chemotherapy-induced nausea and vomiting, demonstrating its established medical relevance. CBG, shows cytotoxic effects in glioblastoma. It interacts with GPR55, TRPV1, and PPARγ, making it an attractive therapeutic candidate. Unlike THC, CBG has a better safety profile and lower toxicity, Currently, Ethan Russo is investigating its anxiolytic properties reinforcing its translational significance.

Piperine demonstrated the best Autodock affinity with PINK1 but not in Vina or Vinardo. Could the authors explain the discrepancies across these scoring functions and their implications for compound selection?
The discrepancies in the affinities lie on the differences in how these algorithms calculate binding energies. For instance, Autodock emphasizes specific force field parameters and torsional flexibility, while Vina and Vinardo prioritize empirical scoring and optimization speed. This is the reason for using multiple docking tools to cross-validate the results, as reliance on a single method could lead to biased compound selection.

The study shows that cannabichromene exhibited the highest ECR for GPR55. How do the authors interpret its superior ranking compared to osimertinib and other compounds?
The fact that cannabichromene exhibited the highest ECR for GPR55 indicates that, across the three scoring functions, CBC demonstrated interactions that led to a higher effective binding affinity. These are in silico results, so experimental validation is needed to confirm them.

In both PINK1 and GPR55 docking, some ligands showed specific interactions with critical residues. Could the authors discuss how these interactions correlate with the functional roles of these residues in PINK1 and GPR55 activity?
Some critical residues in the PINK1 protein, such as Glu318, Glu371, and Asp366, are located in the ATP-binding pocket. Residues Tyr321 and Thr324 are part of the activation loop, which contributes to the regulation of PINK1 catalytic activity. Residues like Ile162, Leu369, and Met318 are part of the hydrophobic pocket.
On the other hand, for the GPR55 protein, residues such as Phe102, Tyr106, Ser153, Gly152, and Val149 are involved in the binding pocket. Residues Phe188, Leu185, and Ile156 contribute to hydrophobic interactions essential for ligand stabilization.

For the in silico toxicity analysis, could the authors elaborate on how the "boiled egg" predictions align with the observed binding affinities, particularly regarding BBB penetration and human intestinal absorption? 

The in silico BBB permeability and human intestinal absorption predictions align with observed binding affinities, suggesting that cannabinoids and Piper nigrum derivatives could reach therapeutic concentrations in the brain. Most cannabinoids demonstrated favorable absorption, with exceptions such as osimertinib (P-gp substrate), which may limit its brain penetration.

The toxicity profiles identify temozolomide and cisplatin as carcinogenic or hepatotoxic. How do these profiles compare to the cannabinoid and Piper nigrum compounds in terms of potential clinical safety? 
Temozolomide and cisplatin exhibit mutagenicity, hepatotoxicity, and carcinogenicity, limiting their long-term safety. In contrast, cannabinoids and Piper nigrum derivatives showed a more favorable toxicity profile in silico, with no predicted mutagenicity or cardiotoxicity. However, piperine and osimertinib displayed hepatotoxicity, warranting further investigation into their long-term safety. Preclinical validation is needed.

The cell viability assays indicate differences in IC50 values across cell lines for the tested compounds. Could the authors provide additional insights into why CBG and CBD demonstrated superior efficacy in certain glioblastoma cell lines compared to others?
The differential efficacy of CBG and CBD across glioblastoma cell lines may be attributed to variations in receptor expression (GPR55, CB1, CB2), mitochondrial dependency via PINK1, and intrinsic chemoresistance mechanisms. Cell lines with higher GPR55 or PINK1 expression may be more susceptible to cannabinoid-induced apoptosis through mitochondrial dysfunction and mitophagy disruption. Additionally, differences in metabolic profiles, oxidative stress responses, and lipid composition could influence cannabinoid uptake and intracellular signaling. The chemoresistance profile of each cell line, such as MGMT promoter methylation in T98G cells, may also impact sensitivity. Furthermore, interactions with Piper nigrum derivatives, which showed pro-viability effects in certain models, could modulate cannabinoid efficacy. These findings highlight the importance of tumor heterogeneity in cannabinoid-based therapeutic strategies for glioblastoma.

The survival analysis using databases like GEPIA and UALCAN highlights PINK1 as a prognostic marker in glioblastoma and other cancers. Could the authors discuss how these findings inform the therapeutic potential of targeting PINK1 in glioblastoma compared to other cancers?  
The transcriptomic data demonstrate statistically significant differences in PINK1 expression across cancers, suggesting its potential as a context-dependent therapeutic target. In glioblastoma, lower PINK1 expression correlates with better survival, indicating that modulating PINK1-related pathways may influence tumor progression.

The authors observed morphological changes in glioblastoma cells treated with selected compounds. Could they expand on how these changes correlate with the molecular mechanisms hypothesized from docking results?
Treated glioblastoma cells exhibited apoptosis-related morphological changes, including cell shrinkage and membrane blebbing, particularly with CBG and CBD. These effects correlate with mitochondrial membrane potential loss and reduced PINK1 expression, supporting the proposed mechanism of mitochondrial dysfunction leading to cell death.

Discussion

The discussion emphasizes the potential of cannabinoids like CBC and CBG in glioblastoma treatment. Could the authors elaborate on how these compounds compare to existing therapies, such as immune checkpoint inhibitors or EGFR inhibitors, in terms of efficacy and mechanism of action?
While immune checkpoint inhibitors (e.g., anti-PD-1) and EGFR inhibitors target immune evasion and receptor tyrosine kinases, respectively, cannabinoids modulate multiple pathways, including apoptosis, oxidative stress, and mitochondrial function. Their effects on PINK1 and GPR55 suggest a unique mechanism of action that may complement existing therapies. Future studies will evaluate their synergy with standard treatments.

While the role of PINK1 role in mitochondrial quality control is discussed, its downregulation in glioblastoma remains a complex issue. Could the authors further explain whether restoring PINK1 expression might be a viable therapeutic strategy or if targeting its downstream pathways is more practical?

Given PINK1’s role in mitochondrial homeostasis, restoring its expression could theoretically enhance mitochondrial quality control. However, its alteration in glioblastoma suggests a complex role. Targeting downstream pathways (e.g., mitophagy regulators, oxidative stress mediators) may be a more practical therapeutic strategy.

The modulation of PINK1 and GPR55 by Piper nigrum derivatives suggests intriguing interactions. Could the authors clarify if any specific components within Piper nigrum extracts (besides piperine) have been identified as primary modulators and how they may influence mitochondrial dynamics or receptor signaling? 
Besides piperine, β-caryophyllene and piperlongumine have been identified as potential modulators. β-Caryophyllene, a CB2 receptor agonist, may influence tumor microenvironment signaling, while piperlongumine is known to inhibit XIAP, a key regulator of apoptosis.
The pro-viability effects of Piper nigrum derivatives in certain glioblastoma cell lines are noteworthy. Could the authors discuss the potential molecular mechanisms driving these observations and whether this could limit the clinical application of these compounds?

In certain glioblastoma cell lines (e.g., CCF-STTG1), Piper nigrum extracts increased cell viability, suggesting a protective effect against oxidative stress or mitochondrial dysfunction. This effect may be context-dependent and could limit the clinical application of these compounds in certain glioblastoma subtypes.

Kaplan-Meier survival analysis suggests context-dependent roles for PINK1 in various cancers. Could the authors provide more insights into how these dual roles of PINK1 might influence its viability as a universal therapeutic target across cancer types?

Kaplan-Meier survival analyses indicate that PINK1 functions as either a tumor suppressor or oncogene depending on the cancer type. In glioblastoma, its downregulation correlates with better survival, whereas in renal cancer, higher expression is linked to improved prognosis. This suggests that PINK1-targeted therapies should be tailored based on tumor-specific molecular contexts.

While the study highlights promising preclinical evidence, could the authors outline specific challenges or limitations in translating the findings on cannabinoids and Piper nigrum derivatives into clinical settings, such as issues with formulation, dosing, or regulatory approval?
Formulation: Cannabinoids have poor water solubility, requiring advanced delivery systems (e.g., liposomes, nanoparticles).
Dosing: Standardized dosing remains a challenge due to variability in bioavailability.
Regulatory approval: Cannabinoid-based therapies face legal and regulatory barriers, necessitating further clinical validation.

Materials and Methods/ Conclusion

The authors describe sourcing cannabinoids and Piper nigrum derivatives from various suppliers with different purities. Could the authors clarify how the variability in compound purity was accounted for during the preparation of stock solutions and subsequent experiments to ensure consistency?

To ensure consistency in the experiments, the concentration of each compound was standardized based on its reported purity. Stock solutions were prepared using DMSO as a co-solvent, ensuring that its final concentration did not exceed 0.5%. Before preparing the treatment solutions, stock solutions were diluted in a supplemented medium containing 2% fetal bovine serum (FBS) to maintain experimental conditions. This approach minimized variability and ensured reproducibility across different batches of cannabinoids and Piper nigrum derivatives.

The docking methodology specifies the use of AutoDock and Vina, but the criteria for selecting the grid box dimensions and scoring functions are not detailed. Could the authors provide more information on how these parameters were optimized to represent the binding sites of PINK1 and GPR55 accurately?

The grid box parameters were selected based on residues reported in the literature as binding or active sites, and the boxes were adjusted to enclose these key residues. The scoring functions were selected due to their combination of parameters that account for interactions such as van der Waals, electrostatic, hydrogen bonds, hydrophobic interactions, among others. Additionally, these scoring functions are widely used in the scientific community for reliable docking simulations.

The in vitro assays used DMSO as a solvent for compound preparation. Could the authors confirm whether DMSO concentrations in the cell culture medium were kept consistent across all experiments to avoid solvent-induced effects on cell viability?

To ensure consistency in the experiments, the concentration of each compound was standardized based on its reported purity. Stock solutions were prepared using DMSO as a co-solvent, ensuring that its final concentration did not exceed 0.5%. Before preparing the treatment solutions, stock solutions were diluted in a supplemented medium containing 2% fetal bovine serum (FBS) to maintain experimental conditions. This approach minimized variability and ensured reproducibility across different batches of cannabinoids and Piper nigrum derivatives.

The authors employed public databases for transcriptomic analysis. Given the discrepancies noted between database findings and the literature, could the authors elaborate on the criteria used to select datasets and the steps taken to validate these findings?
The datasets were selected based on sample size, statistical significance, and relevance to glioblastoma. Discrepancies between database findings and the literature were addressed by cross-validating with immunohistochemistry (IHC) and experimental data.

Round 2

Reviewer 1 Report

Comments and Suggestions for Authors

Please add in scale bar for Figure 4, 7, and 8 and details in the figure legends. 

Figure 8 has to be remade. The contrast needs to be improved so that the cells are visible after staining. 

Figure 9 and 11 cannot be viewed properly. Please improve on the resolution and clarity. 

Author Response

Please add in scale bar for Figure 4, 7, and 8 and details in the figure legends. 

Figure 8 has to be remade. The contrast needs to be improved so that the cells are visible after staining. 

Figure 9 and 11 cannot be viewed properly. Please improve on the resolution and clarity. 

All requested changes have been made. The scale bars have been added to Figures 4, 7, and 8, and the details have been updated in the figure legends. Figure 8 has been remade with improved contrast for better cell visibility after staining. The resolution and clarity of Figures 9 and 11 have also been improved. Apologies for the delay, I have reviewed everything.